# Novel index to comprehensively evaluate air cleanliness: the "Clean aIr Index"

Tomohiro O. Sato[1], Takeshi Kuroda[2,1], and Yasuko Kasai[1,3]

[1]National Institute of Information and Communications Technology
[2]Tohoku University
[3]University of Tsukuba

**Correspondence:** Yasuko Kasai (ykasai@nict.go.jp)

**Abstract.** Air quality on our planet has been changing in particular since the industrial revolution (1750s) because of anthropogenic emissions. It is becoming increasingly important to visualize air cleanliness, since clean air is as valuable a resource as clean water. A global standard to quantify the level of air cleanliness is swiftly required, and we defined a novel concept, namely "Clean aIr Index, CII." The CII is a simple index defined by the normalization of the amount of a set of individual air pollutants. A CII value of 1 indicates completely clean air (no air pollutants), and 0 indicates the presence of air pollutants up to numerical environmental criteria for the normalization. In this time, the air pollutants used in the CII were taken from the Air Quality Guidelines (AQG) set by the World Health Organization (WHO), namely $O_3$, particulate matters, $NO_2$ and $SO_2$. We chose Japan as a study area to evaluate CII because of the following reasons: i) accurate validation data, as the in situ observation sites of the Atmospheric Environmental Regional Observation System (AEROS) provide highly accurate values of air pollutant amounts, ii) fixed numerical criteria, the Japanese Environmental Quality Standards, as directed by the Ministry of the Environment (MOE). We quantified air cleanliness in terms of the CII for the all 1896 municipalities in Japan, and used data from Seoul and Beijing to evaluate Japanese air cleanliness. The amount of each air pollutant was calculated using a model that combined the Weather Research and Forecasting (WRF) and Community Multiscale Air Quality (CMAQ) models for 1 April 2014 to 31 March 2017. The CII values calculated by the WRF-CMAQ model and the AEROS measurements showed good agreement. The mean of correlation coefficient for the CII values of 498 municipalities where the AEROS measurements have operated was 0.66±0.05, which was higher than that of Air Quality Index (AQI) of 0.57±0.06. The CII values averaged for the study period was 0.67, 0.52 and 0.24 in Tokyo, Seoul and Beijing, respectively, thus, the air in Tokyo was 1.5 and 2.3 times cleaner, i.e., less amounts of air pollutants, than those in Seoul and Beijing, respectively. The average CII value for the all Japanese municipalities was 0.72 over the study period. The extremely clean air, CII ≈ 0.90, occurred in southern remote islands of Tokyo and around west of the Pacific coast, i.e., Kochi, Mie and Wakayama Prefectures during summer with transport of clean air from the ocean. We presented "Top 100 clean air cities" in Japan as one example of application using CII in society. We confirmed that the CII enabled the quantitative evaluation of air cleanliness. The CII can be useful value in various scenarios, such as encouraging sightseeing and migration, investment and insurance company business, and city planning. The CII is a simple and fair index that can be applied to all nations.

# 1   Introduction

Air is an essential components for all life on our planet. Air quality has been changing since the industrial revolution (1750s). According to the report from OECD (2016), air pollutant emissions are predicted to increase because of the projected increase in the energy demand, e.g., transportation and power generation using fossil fuels, especially in East Asia. This report also mentions that the global annual market costs are predicted to increase from $0.3\%$ in 2015 to $1.0\%$ in 2060 of global GDP because of reduced labor productivity, increased health expenditures, and crop yield losses due to air pollution.

A global standard index to quantify air cleanliness should be developed as the Global Drinking Water Quality Index (GDWQI), for water quality, defined by UNEP (2007), since clean air is as valuable a resource as clean water is. Such an index can be a useful communication tool to help decision making. The index should be understandable/informative not only for scientific experts but also general citizen, and also be upgraded with the scientific data.

Several indices exist for estimating air quality, e.g., Air Quality Index (AQI) in the United States (US EPA, 2006) and Air Quality Health Index in Canada (Stieb et al., 2008) and Hong Kong (Wong et al., 2013). The purpose of these indices is to estimate health risks due to air pollution exposure. These indices were developed based on epidemiological studies and optimized for each country or local area. The most commonly used index is the US AQI (US EPA, 2006). The AQI ranges from 0 to 500 and is calculated based on the concentrations of the six air pollutants. In the calculation of AQI, an individual AQI for every air pollutants are calculated for a given location on a given day, and the maximum of all individual AQIs is defined as the overall AQI. Hu et al. (2015) performed a comparison study of several indices for air quality using the measurements in China, and showed AQI underestimates the severity of the health risk associated with the exposure to multi-pollutant air pollution because AQI does not appropriately represent the combined effects of exposure to multiple pollutants. An index to quantify the air quality is still under development, and the global standard has not been established yet.

In this study, we propose a novel concept of index to quantify air cleanliness, "Clean aIr Index (CII)" to establish the global standard for quantifying air cleanliness. The purpose of CII is to comprehensively evaluate air cleanliness by normalizing the amounts of common air pollutants with numerical environmental criteria. In this time, we selected surface $O_3$, particulate matter (PM), $NO_2$, and $SO_2$ from the Air Quality Guidelines (AQG) set by the World Health Organization (WHO)(WHO, 2005). As a first approach, we chose Japan for evaluating the CII because of i) the validation data, as the in situ observation sites of the Atmospheric Environmental Regional Observation System (AEROS) provide highly accurate air pollutant amounts, and ii) the fixed numerical criteria, i.e., the Japanese Environmental Quality Standards given by the Ministry of the Environment (MOE).

In this paper, Sect. 2 defines the CII. Section 3 describes the model for calculating the CII for all Japanese municipalities, and validates the CII values by comparing with those derived from AEROS measurements. In Sect. 4, air cleanliness in each municipality is quantified and the area and season of high air cleanliness in Japan is identified using the CII.

**Table 1.** Value of numerical criteria ($s$), $O_3$, suspended particulate matter (SPM), $NO_2$, and $SO_2$ used in this study. We used the criteria of the Japanese Environmental Quality Standards (JEQS) given by the Ministry of the Environment (MOE) of Japan. Average of air pollutant amount calculated by the model for all Japanese municipalities over the study period is shown.

| Air pollutant | Average of model | Numerical criteria ($s$) | Notes |
|---|---|---|---|
| $O_3$ | 46.4 ppb | 60 ppb | Threshold of the hourly values |
| SPM | 13.5 $\mu$g/m$^3$ | 100 $\mu$g/m$^3$ | Threshold of the daily average for hourly values |
| $NO_2$ | 10.5 ppb | 60 ppb | Threshold of the daily average for hourly values |
| $SO_2$ | 1.9 ppb | 40 ppb | Threshold of the daily average for hourly values |

## 2 Clean aIr Index (CII)

The CII is a simple index defined by the normalization of each air pollutant amount. The definition of CII is given by

$$\text{CII} = f(x, s) = 1 - \frac{1}{N} \sum_{i}^{N} \frac{x[i]}{s[i]}, \tag{1}$$

where $x[i]$ is the amount of $i$th air pollutant, $s[i]$ is the numerical criteria for the normalization of $x[i]$, and N is the number of air pollutants considered in the CII. In this equation, a higher CII value indicates cleaner air, with a maximum of 1 indicating the absence of air pollutants. The CII value decreases as the amount of air pollutant increases, with a value of 0 indicating that the amount of air pollutant is equal to the numerical criteria and a negative value indicating that the amount of air pollutant is larger than the numerical criteria.

In this study, the air pollutants used in the CII are $O_3$, PM, $NO_2$ and $SO_2$ following the WHO AQG (WHO, 2005) as mentioned above, i.e., N = 4. The field of this study is Japan, thus, we set the values of $s$ according to the Japanese Environmental Quality Standards (JEQS), which are given by the Ministry of the Environment (MOE) of Japan (Table 1). The time length should be consistent between the $x$ and $s$ values to implement the air pollutant amount in the calculation of CII. In this case, the $s$ value of $O_3$ is defined as a threshold for 1-hour average, and those of the others are defined as 24-hour average. We employed the maximum of 1-hour average per day for $O_3$ and daily-mean for the other pollutants. We used the criterion for photochemical oxidants (Ox) in the JEQS as the $s$ value for $O_3$, because more than 90–95 % of Ox is composed of $O_3$ (Akimoto, 2017). The CII can be used both globally and locally by defining the setting of $s$ values. In case of applying the CII to compare the air cleanliness globally, the numerical criteria should be given by the WHO AQG (WHO, 2005).

The selected air pollutants have been of importance for the last 5 decades in Japan, and have been monitored by AEROS from 1970. Surface $O_3$, which is harmful to human health (e.g., Liu et al., 2013) and crop yields and quality (e.g., Feng et al., 2015; Miao et al., 2017), has been increasing in Japan since the 1980s in spite of the decreasing $O_3$ precursors, such as $NO_X$ and volatile organic compounds (VOCs) (Akimoto et al., 2015). Nagashima et al. (2017) estimated that the source of surface $O_3$ is increasing, and approximately 50 % of the total increase was caused by transboundary pollution from China and Korea. We used the suspended particulate matter (SPM) for PM following the JEQS. $NO_2$ is a precursor of surface $O_3$ and is a harmful

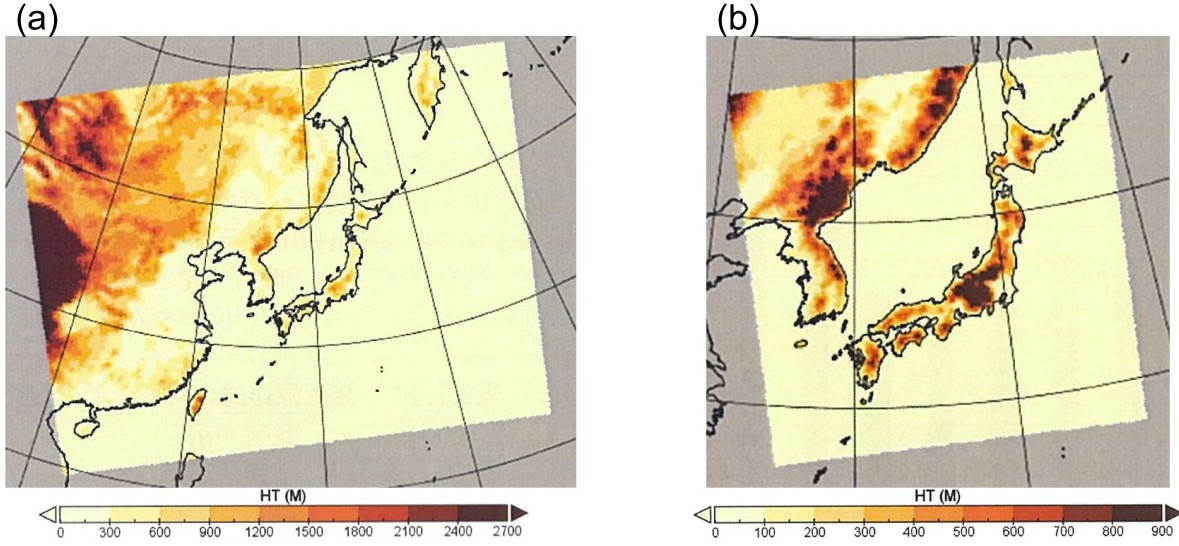

**Figure 1.** Ranges of (a) Domain 1 and (b) Domain 2 of the WRF-CMAQ models in this study. Color bars denote altitude.

pollutant. It mostly originates from anthropogenic sources, especially fossil fuel combustion (e.g., power plants and vehicles). The environmental $SO_2$ level was severe in 1970s in Japan. But the $SO_2$ concentration has been decreasing owing to the

use of desulfurization technologies and low-sulfur heavy oil, and JEQS for $SO_2$ was satisfied at most AEROS sites in 2012 (Wakamatsu et al., 2013).

## 3  Model simulation

A model simulation was performed to calculate the amounts of $O_3$, SPM, $NO_2$, and $SO_2$ of all Japanese municipalities (1896 in total; note that wards in megacities, such as Tokyo, Osaka, and Fukuoka were counted as independent municipalities). The

AEROS measurement network does not cover the all municipalities, thus we employed the model simulation. We combined two regional models; The Weather Research and Forecasting (WRF) model, for calculating meteorological fields (e.g., temperature, wind, and humidity), and the Community Multiscale Air Quality (CMAQ) model, calculating air pollutant amounts using the WRF results as input parameters. Detailed descriptions about the WRF and CMAQ models are written in Sect. 3.1. The calculations were made from 22 March 2014 to 31 March 2017, and the outputs from 1 April 2014 to 31 March 2017 with the

interval of every 1 hour were used for analyses. We selected the simulation period with a unit of fiscal year (FY), starting on 1 April and ending on 31 March, because the AEROS measurement dataset that we used to evaluate our simulation (Sect. 3.3) was archived with a unit of FY. The amount of SPM was simply assumed as $[SPM] = ([PM_{10}] + [PM_{2.5}])/2$ in this study.

## 3.1 WRF-CMAQ settings

We used the WRF model version 3.7 (Skamarock et al., 2008) to calculate the meteorological fields. We set two model domains; which Domain 1 covered East Asia with a horizontal grid resolution of 40 km and $157 \times 123$ grid points, and Domain 2 covered main-land Japan with a horizontal grid resolution of 20 km and $123 \times 123$ grid points, see Fig. 1. The vertical layers consisted of 29 levels from the surface to 100 hPa. The initial and boundary conditions were obtained from the National Center for Environmental Prediction (NCEP) Final Operational Global Analysis (FNL, ds083.2) data (six-hourly, $1° \times 1°$ resolution) (NCEP FNL, 2000). In the model domain, three-dimensional grid nudging for horizontal wind, temperature, and water vapor mixing ratio as well as two-dimensional grid nudging for sea surface temperature were performed every six hours. Furthermore, we used the following parameterizations: the new Thompson scheme (Thompson et al., 2008) for microphysical parameterization, the Dudhia scheme (Dudhia, 1989) and Rapid Radiative Transfer Model (Mlawer et al., 1997) for short- and longwave radiation processes, the Mellor-Yamada-Janjić scheme (Janjić, 1994) for planetary boundary layer parameterization, and the Betts-Yamada-Janjić scheme (Janjić, 1994) for cumulus parameterization.

The CMAQ model version 5.1 was used as a chemical transport model in this study. Byun and Schere (2006) provided an overview of the CMAQ model, and the updates and scientific evaluations of CMAQ version 5.1 are provided by Appel et al. (2017). For the gas-phase chemistry, the 2005 Carbon Bond (CB05) chemical mechanism with toluene update and additional chlorine chemistry (CB05TUCL Yarwood et al., 2005; Whitten et al., 2010; Sarwar et al., 2012) was used. The core CB05 mechanism (Yarwood et al., 2005) has 51 chemical species and 156 reactions for the compounds and radicals of hydrogen, oxygen, carbon, nitrogen and sulfur. After that, the toluene update (Whitten et al., 2010) has improved the predictions of $O_3$ and $NO_X$ productions and losses dealing with 59 chemical species and 172 reactions in total. In addition, the implementation of chlorine chemistry (Sarwar et al., 2012) added 7 chemical species and 25 reactions of chlorides, affecting to increase $O_3$ and reduce nitrates. About the photolysis of molecules, the photolysis rate preprocessor (JPROC) with 21 reactions (Roselle et al., 1999) has been implemented. About the formations of aerosols, the combination of secondary organic aerosol (SOA) formations (Pye and Pouliot, 2012; Pye et al., 2013; Appel et al., 2017), ISORROPIA algorithms (Fountoukis and Nenes, 2007) and binary nucleation (VehkamäKi et al., 2002) has been implemented. 45 kinds of aerosols components, including sulfate, ammonium, black carbon, organic carbon and sea salt, have been considered in this model.

The molecules and aerosols were provided by the emissions (anthropogenic, biogenic and sea salt) from surface or transports from the boundaries of domains, and were transported by the wind fields calculated in the WRF model and the parameterizations of horizontal/vertical diffusions, dry deposition and gravitational settling (see Byun and Schere, 2006; Appel et al., 2017). Anthropogenic emissions were defined using the MIX Asian emission inventory version 1.1 which included emissions by power, industry, residential, transportation and agriculture (Li et al., 2017). This inventory of $SO_2$, $NO_X$, PM, VOC, CO and $NH_3$ for 2015 were estimated by correcting the 2010 data (2008 for $NH_3$) and implemented into the CMAQ model. The corrections were made using the statistical secular changes in the annual total anthropogenic emissions of pollutants and $CO_2$ (Crippa et al., 2019), population, amount of used chemical fertilizer and $NH_3$ emission by farm animals for each country included in the model domains (Japan, China, South Korea, North Korea, Taiwan, Mongolia, Vietnam, and Far East Russia). Biogenic

emissions of VOC were provided by the Model of Emissions of Gases and Aerosols from Nature (MEGAN) version 2.10 (Guenther et al., 2012) using the meteorological fields calculated by the WRF model for 2016. Those implemented emission inventories did not include interannual changes. Volcanic emissions of $SO_2$ were ignored in our model simulation because of the following reason. The $SO_2$ concentration values were averaged for 24 hours to be consistent with the time length of the numerical criterion of JEQS. This procedure dilutes an increase of $SO_2$ due to volcanic eruption.

The CMAQ model used two model domains, whose regions were the same as those adopted in the WRF model, see Fig. 1, and vertical coordinates of 22 layers; the thickness of the lowest layer was approximately 30 m. The initial and boundary conditions of air pollutants for Domain 1 were obtained from the Model for OZone And Related chemical Tracers (MOZART) version 4 (Emmons et al., 2010), and the boundary conditions for Domain 2 were the model outputs of Domain 1. The MOZART provided the distributions of more than 80 kinds of chemical species and aerosols for the inputs of our model calculations. The amount of pollutants in each Japanese municipality were defined at the longitude/latitude of the municipal office, with the weighted average of the outputs at model grid points near the municipal office using the following equation:

$$\overline{A} = \frac{1}{A_w} \sum_{i=1}^{I} \frac{R^2 - d_i^2}{R^2 + d_i^2} A_i, \quad A_w = \sum_{i=1}^{I} \frac{R^2 - d_i^2}{R^2 + d_i^2}, \tag{2}$$

where $\overline{A}$ is the defined amount of a pollutant at the municipal office, $I$ (=2 or 3 mostly) is the number of the model grid points of Domain 2 within $R = \sqrt{0.02}$ degrees of the terrestrial central angle (approximately 16 km) from the office, and $A_i$ and $d_i$ are the simulated amount of a pollutant and distance from the office, respectively, at each model grid point. Note that Okinawa Prefecture and Ogasawara-mura municipality in Tokyo Prefecture were outside Domain 2, and the amount of pollutants at the municipalities in them were thus defined using the model outputs of Domain 1 with $R = \sqrt{0.08}$ degrees (approximately 31 km) in Eq. (2). We also derived the amount of pollutants in Seoul and Beijing for the comparisons with that inside Japan from the model outputs of Domain 1.

### 3.2 Spatial-temporal variation of CII

The spatial-temporal variations of CII based on the WRF-CMAQ model are shown in Fig. 2 (a). The horizontal and vertical axes correspond to the date and municipal number, respectively. The lower municipal number corresponds approximately to the municipalities in northeast Japan and vice versa, and the major cites in Japan are shown in the vertical axis. The CII value clearly depended on both area and season. The CII value tended to be higher in summer because of transportation of unpolluted air mass from the Pacific Ocean. In August 2014, July 2015 and September 2016, the CII values of almost all municipalities were higher than 0.9 for a few weeks. However, the local CII values decreased to below 0.5 over a short period from May because of local air pollutant emissions and the enhancement due to photochemical reactions induced by strong UV sunlight. The CII value was moderate (0.7–0.8) and stable from November to February over Japan but gradually decreased from February to May or June because polluted air was transported from East Asia (e.g., Park et al., 2014), and the sunlight strengthened.

These spatial-temporal features were reproduced by the AEROS measurements. Figures 2 (b) and (c) show the time series variations in the daily CII value derived from the AEROS measurements and the difference (WRF-CMAQ − AEROS), respec-

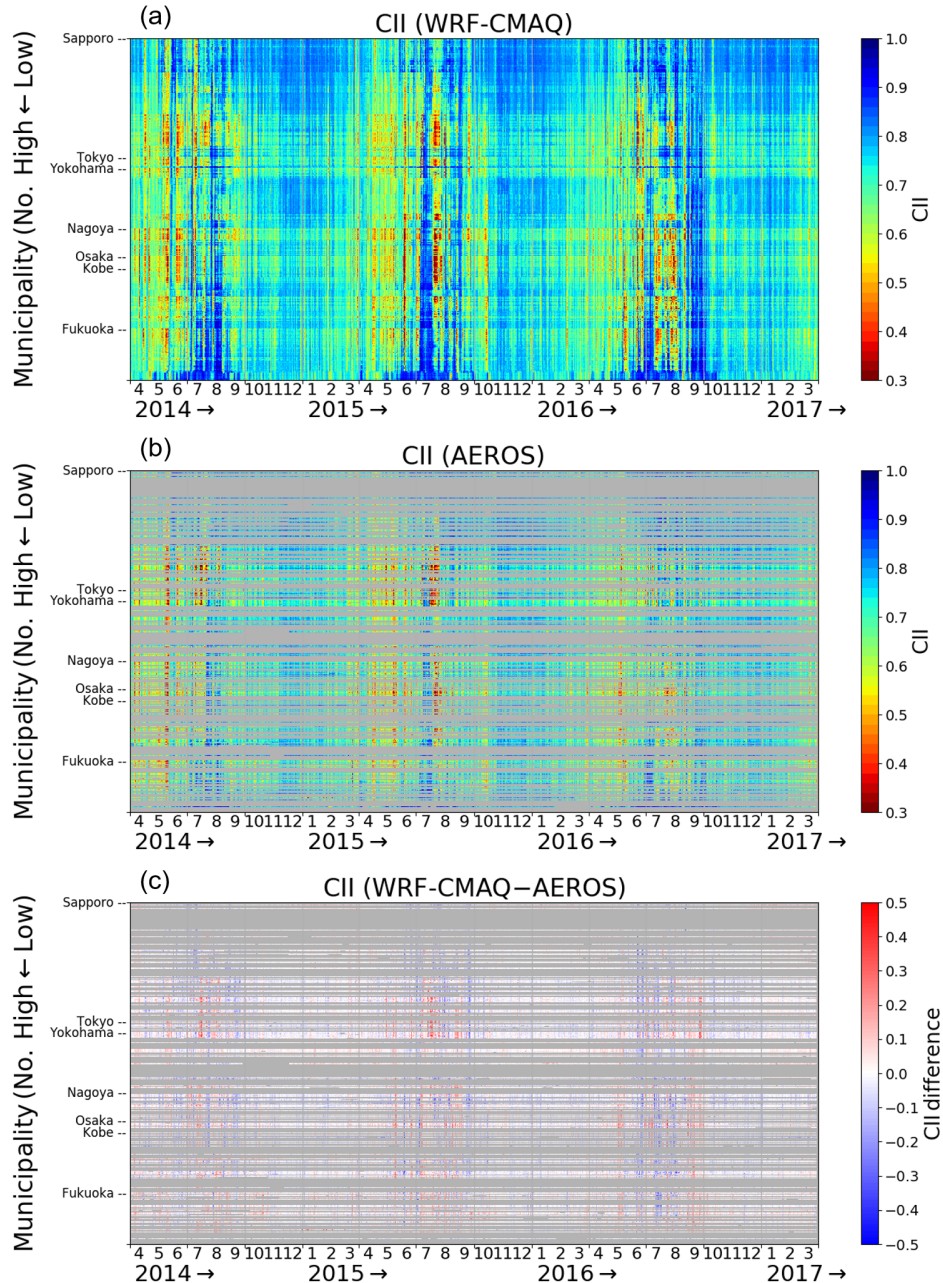

**Figure 2.** Spatial-temporal variation in CII values derived from (a) the WRF-CMAQ model, (b) the AEROS measurements and (c) their difference (WRF-CMAQ − AEROS). The horizontal and vertical axis corresponds to date of the study period and Japanese municipal number, respectively. The municipalities where the AEROS observation covers less than 20 % of days in the study period are masked by gray color.

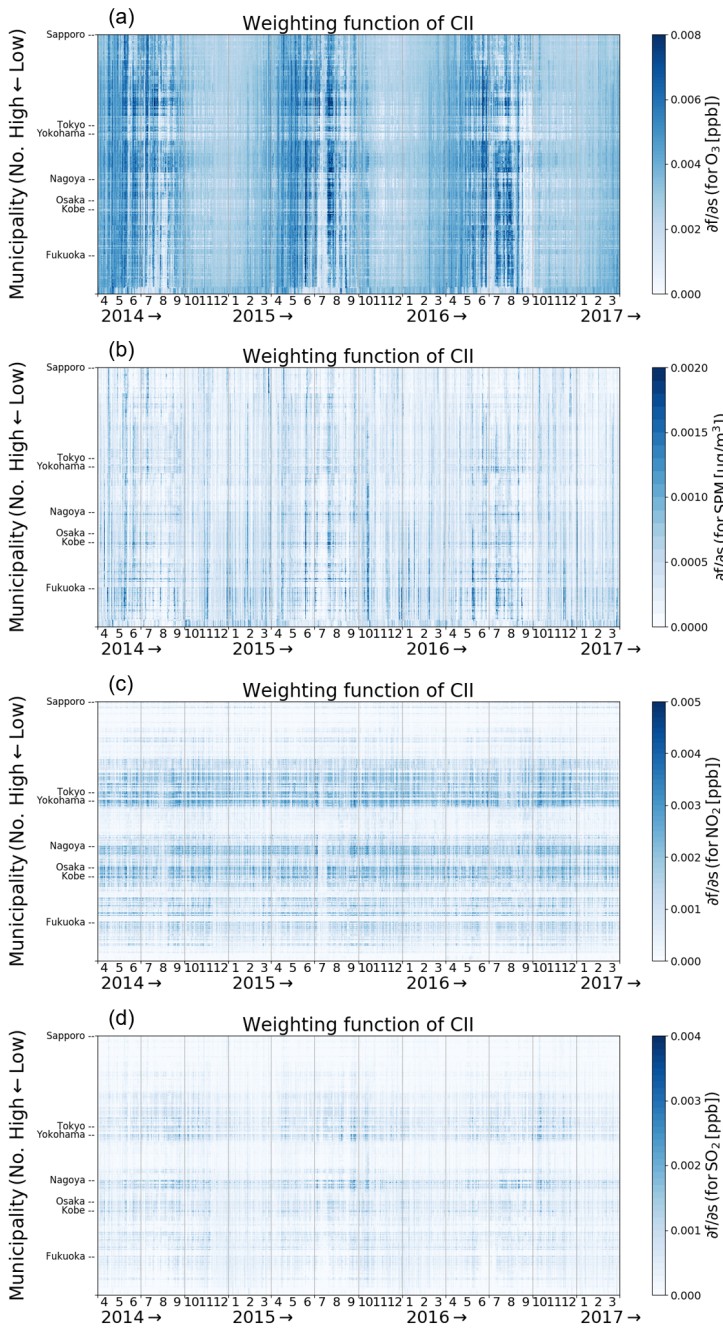

**Figure 3.** Spatial-temporal variation of the weighting function for the numerical criteria, $K_s$, for (a) O$_3$, (b) SPM, (c) NO$_2$ and (d) SO$_2$ derived from the WRF-CMAQ model. The color scaling is optimized for each panel.

tively. AEROS is operated by the MOE of Japan and has 1901 observation sites for monitoring air pollutants in FY2016. The AEROS data were obtained from the atmospheric environment database of the National Institute for Environmental Studies (*Kankyosuchi database (in Japanese)*). We used the AEROS observation sites that cover more than 80 % of days in the study period, and 498 in 1896 municipalities were covered by the AEROS measurements. The AEROS measurement results were averaged for all observation sites in each municipality in case that there were several observation sites in one municipality. In this comparison, the AEROS Ox data were compared to the WRF-CMAQ $O_3$ data because the composition ratio was larger than 90–95 % $O_3$ in Ox (Akimoto, 2017).

The CII value depends not only on the amount of $O_3$, SPM, $NO_2$, and $SO_2$ ($x$), but also on their numerical criteria ($s$), see Eq. (1). A partial differentiation analysis was performed to determine the sensitivities of the $s$ values of $O_3$, SPM, $NO_2$, and $SO_2$ to CII. Figure 3 shows the weighting function for the numerical criteria ($K_s$) given by

$$K_s[i] = \frac{\partial f(x,s)}{\partial s[i]} = \frac{1}{N} \frac{x[i]}{s[i]^2}. \tag{3}$$

As shown in Eq. (3), $K_s$ positively correlates with $x$, and the CII value monotonically increases with increasing $s$. The temporal variation in CII primarily corresponded with the variation in $O_3$. The average $K_s$ for $O_3$ was highest among the species used to calculate the CII in this study, because the $x/s$ value of $O_3$ was higher than those of SPM, $NO_2$, and $SO_2$ (Table 1). The value of $K_s$ for SPM in western Japan was higher than that in eastern Japan during winter and spring because of the effect of transboundary pollution from East Asia (e.g., Park et al., 2014). The spatial distribution of CII corresponded to those of $K_s$ for $NO_2$ and $SO_2$, which explicitly reflected local emission sources, such as megacities and industrial areas. Typical lifetime of $NO_2$ is approximately a few hours (e.g., Kenagy et al., 2018), and the transport effect was therefore less for these species. We ignored $SO_2$ emissions from volcanic eruptions, and the $SO_2$ distribution consequently corresponded to industrial activities. The spatial distribution of $O_3$ was negatively correlated to that of $NO_2$ primarily because of the reactions (R1-R3).

### 3.3   Evaluation of spatial and temporal bias

We discuss the spatial and temporal bias in our calculation to clarify magnitude of significant differences in the CII value. We compared the CII mean of all days in the study period between WRF-CMAQ and AEROS for each municipality. The CII difference for each municipality is shown in Fig. 4 (a) to investigate the spatial bias. The histogram of the CII difference showed an asymmetric distribution, thus we fitted the histogram by using the Johnson SU function, which is a probability distribution transformed from the Normal distribution to cover the asymmetry of the sample distribution (Johnson, 1949). The mean and standard deviation ($1\sigma$) of CII difference were 0.00 and 0.02, respectively. In the similar way, we compared the CII mean of all Japanese municipalities for each day during the study period to investigate the daily temporal bias. Figure 4 (b) shows the histogram of the CII difference for each day. The mean and standard deviation ($1\sigma$) of CII difference were 0.00 and 0.04, respectively. Hereafter, we average the CII values for at least 30 days to compare the CII value among municipalities to reduce the temporal bias to be less than 0.01 ($\approx 0.04/\sqrt{30}$). Consequently, the difference in CII derived from the WRF-CMAQ larger than 0.02 was significant to be reproduced by AEROS by averaging 30 values.

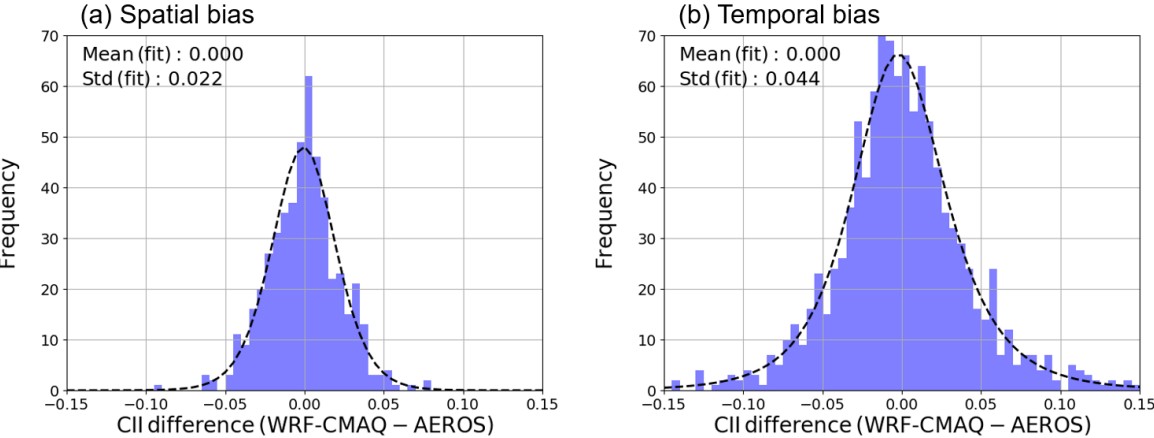

**Figure 4.** Histogram of CII difference between the WRF-CMAQ model and the AEROS measurements. (a) The CII mean values of all days in the study period are compared for each municipality. (b) The CII mean values of all Japanese municipalities are compared for each day. The dashed line represents fitting of the histogram of CII difference by the Johnson SU function. The mean and standard deviation ($1\sigma$) values of the fitting function are shown in the upper left.

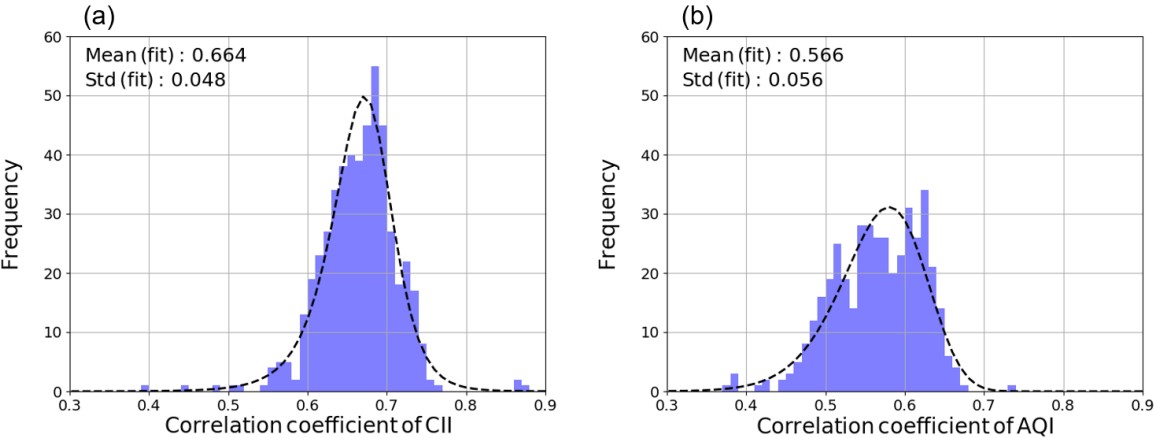

**Figure 5.** Histogram of correlation coefficient ($r$) of municipal mean of daily (a) CII and (b) AQI values for the study period between the WRF-CMAQ model and the AEROS measurements. The dashed line represents fitting of the histogram of CII difference by the Johnson SU function. The mean and standard deviation ($1\sigma$) values of the fitting function are shown in the upper left.

## 3.4 Comparison of CII and AQI

In Sect. 3.4, we discuss the difference between CII and AQI as representative of indices to evaluate the air quality. We compared these indices calculated from the WRF-CMAQ model and the AEROS measurements. The mean of the correlation coefficient ($r$) for the study period between WRF-CMAQ and AEROS was calculated for each municipality. Figure 5 shows the histogram of $r$ for all municipalities for (a) CII and (b) AQI. The histogram was fitted by the Johnson SU function. The $r$ of CII and AQI was $0.66\pm0.05\,(1\,\sigma)$ and $0.57\pm0.06\,(1\,\sigma)$, respectively, and the CII showed better agreement between WRF-CMAQ and AEROS than AQI.

This discrepancy between CII and AQI is explained by the difference of their definitions. In the definition of AQI, only the air pollutants that cause the largest health risk are taken into account and the other air pollutants are ignored (US EPA, 2006). In the definition of CII, four air pollutants, $O_3$, SPM, $NO_2$ and $SO_2$, are averaged with normalization by their numerical criteria, as Eq. (1). It was reported that the amount of the surface $O_3$ was overestimated by the CMAQ model (Akimoto et al., 2019). In this case, $NO_2$ is underestimated because of the following reactions:

$$NO_2 + h\nu \rightarrow NO + O, \tag{R1}$$

$$O + O_2 + M \rightarrow O_3 + M, \tag{R2}$$

$$NO + O_3 \rightarrow NO_2 + O_2, \tag{R3}$$

where M is a third body for the ozone formation reaction. This discrepancy is less affected for CII than for AQI because the amounts of air pollutants are averaged by being normalized by the numerical criteria.

## 4 Visualization of air cleanliness in Japan

In Sect. 4, we discuss the area and season of high air cleanliness in Japan. Figure 6 shows the average CII over the study period (FY2014–2016) for each Japanese municipality. The average CII of 85 % of municipalities were higher than that of Tokyo (23 wards), and those of all the municipalities were higher than those of Seoul and Beijing. Here the JEQS values were employed to the $s$ values to calculate the CII values in Seoul and Beijing to directly compare with those in Japanese municipalities. The average and standard deviation $(1\,\sigma)$ of CII was $0.67\pm0.10$, $0.52\pm0.18$, and $0.24\pm0.32$ in Tokyo, Seoul, and Beijing, respectively. The value of $1 - $ CII monotonically increases with air pollutant amounts increase, and the air in Tokyo was 1.5 and 2.3 times cleaner, i.e., less air pollutant amounts, than those in Seoul and Beijing, respectively. The location of the municipalities discussed hereafter is shown in Fig. 7.

### 4.1 Area and season of high air cleanliness

We discuss the area and season of highest air cleanliness over Japan using the CII in Sect. 4.1. First, the CII average over the study period, FY2014–2016, in each municipality was compared in Fig. 8 (a). The CII averages in northern Japan were higher, and those in municipalities around megacities and industrial areas were lower than the average of all municipalities, $0.72\pm0.04$

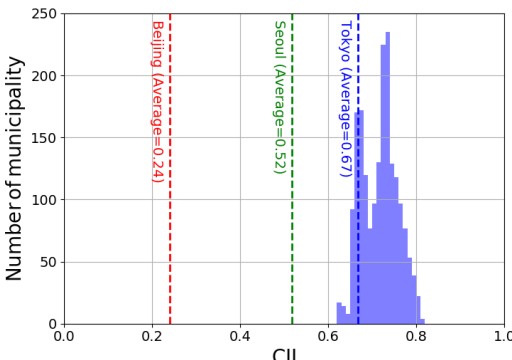

**Figure 6.** Histogram of average CII over the study period (FY2014–2016) for each municipality in Japan. Red, green, and blue dashed lines represent average CII of Beijing, Seoul, and Tokyo (23 wards), respectively.

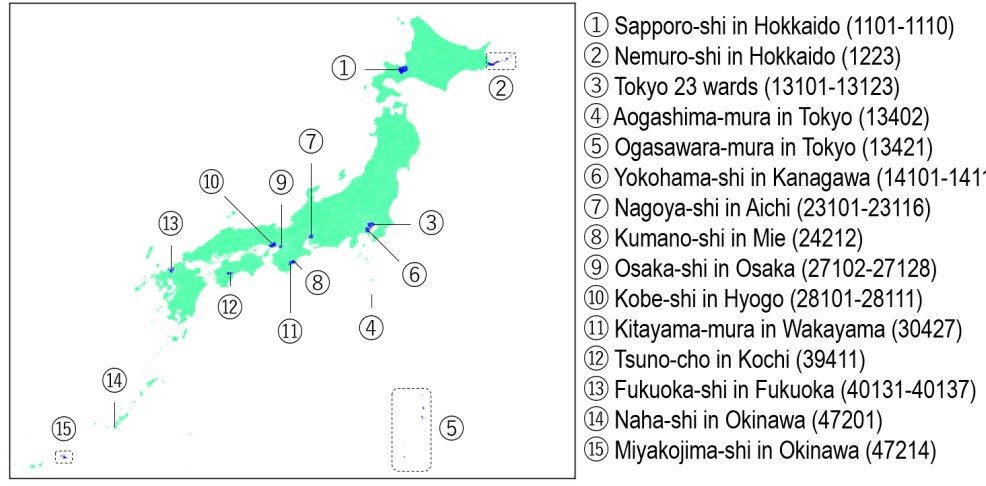

① Sapporo-shi in Hokkaido (1101-1110)
② Nemuro-shi in Hokkaido (1223)
③ Tokyo 23 wards (13101-13123)
④ Aogashima-mura in Tokyo (13402)
⑤ Ogasawara-mura in Tokyo (13421)
⑥ Yokohama-shi in Kanagawa (14101-14118)
⑦ Nagoya-shi in Aichi (23101-23116)
⑧ Kumano-shi in Mie (24212)
⑨ Osaka-shi in Osaka (27102-27128)
⑩ Kobe-shi in Hyogo (28101-28111)
⑪ Kitayama-mura in Wakayama (30427)
⑫ Tsuno-cho in Kochi (39411)
⑬ Fukuoka-shi in Fukuoka (40131-40137)
⑭ Naha-shi in Okinawa (47201)
⑮ Miyakojima-shi in Okinawa (47214)

**Figure 7.** Location of Japanese municipalities focused on in this study. The municipal number is shown in parenthesis.

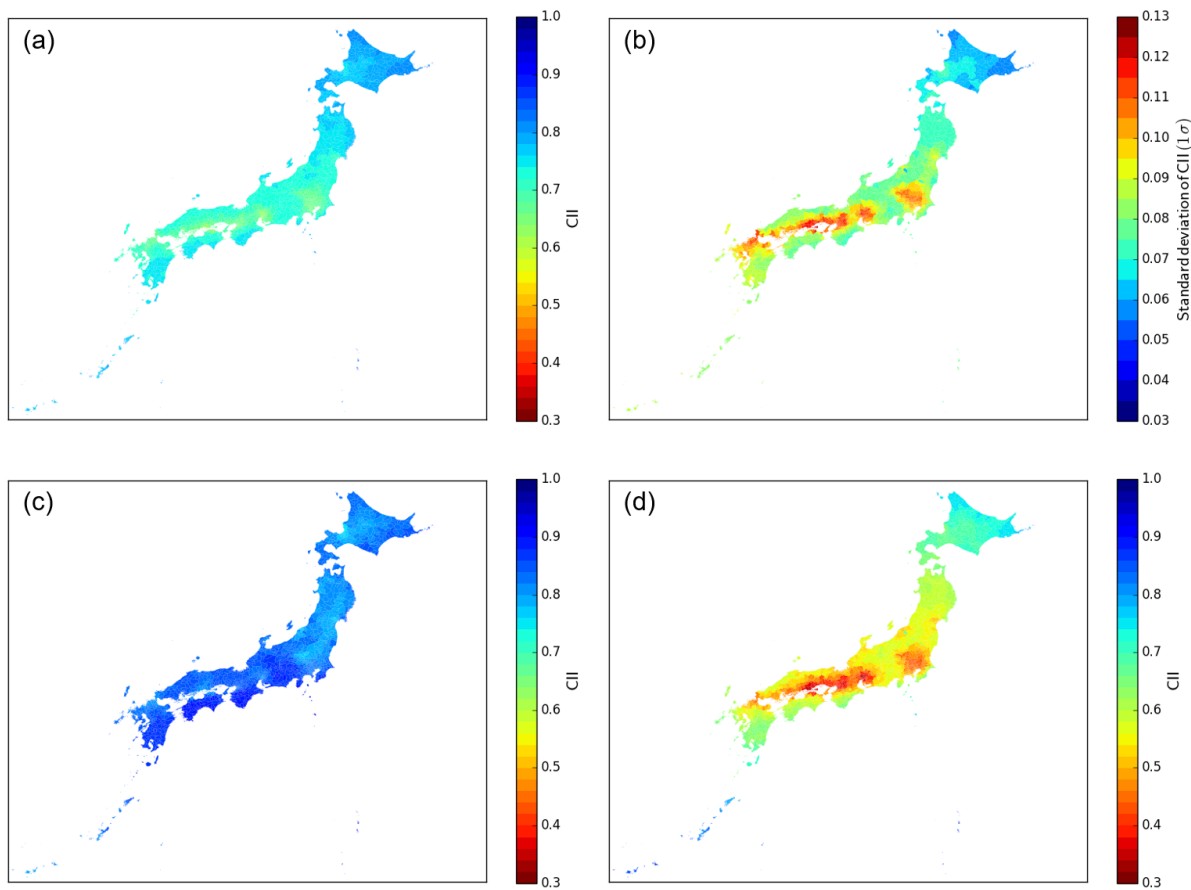

**Figure 8.** Spatial distributions of CII derived from the WRF-CMAQ model. (a) Mean over the study period (FY2014–2016). (b) Same as (a) but for standard deviation ($1\sigma$). (c) Mean for 30 days of highest CII average in all Japanese municipalities (10 days from each FY). (d) Same as (c) but for lowest CII average.

**Table 2.** Ten municipalities with highest average CII value over the study period, FY2014–2016. The municipal number is shown in parenthesis.

| Municipality | Prefecture | CII |
|---|---|---|
| Nemuro-shi (1223) | Hokkaido | 0.814 |
| Hamanaka-cho (1663) | Hokkaido | 0.813 |
| Akkeshi-cho (1662) | Hokkaido | 0.812 |
| Betsukai-cho (1691) | Hokkaido | 0.812 |
| Nakashibetsu-cho (1692) | Hokkaido | 0.809 |
| Kushiro-cho (1661) | Hokkaido | 0.809 |
| Rausu-cho (1694) | Hokkaido | 0.808 |
| Shibetsu-cho (1693) | Hokkaido | 0.808 |
| Ogasawara-mura (13421) | Tokyo | 0.808 |
| Sarufutsu-mura (1511) | Hokkaido | 0.808 |
| Average of all Japanese municipalites | | 0.717 |

($1\,\sigma$). Table 2 shows the 10 municipalities with the highest average CII values, which located in eastern Hokkaido and southern remote island in Tokyo. The average CII was approximately 0.81 in these 10 municipalities and the standard deviation ($1\,\sigma$) over the study period was lower than that in other areas, see Fig. 8 (b), which means the CII remained high throughout the year. For example, the CII daily value in Nemuro-shi municipality, where the three-year CII average was the highest, was higher than the total municipal average of 0.72 in 95 % of days over the study period.

We discuss the CII distribution in case of high CII average of all Japanese municipalities. We selected 10 days per year, a total of 30 days with the highest average CII values (7/9, 7/10, 8/9, 8/10, 8/15, 8/16, 8/18, 10/5, 12/6, 1/12 in FY2014; 7/9, 7/13, 7/16–19, 7/22, 8/17, 9/9, 9/10 in FY2015; and 7/9, 9/7, 9/12–9/14, 9/19, 9/20, 9/25, 9/27, 9/28 in FY2016). The 30 days CII values were averaged to discuss the CII distribution by same-order precision of 0.02 with the AEROS measurements, see Sect. 3.3. Almost all of these 30 days were in summer when unpolluted air was transported from the Pacific Ocean. The average CII values on these 30 days for each municipality are displayed in Fig. 8 (c), and Table 3 shows the 10 municipalities with the highest average CII values on these days. These 10 municipalities located in southern remote islands of Tokyo and western Pacific coast area, i.e., Kochi, Mie and Wakayama Prefectures. The average CII of Aogashima-mura municipality in southern remote islands of Tokyo Prefecture was the highest. The average CII of these 10 municipalities was approximately 0.90, which was 0.06, by CII, higher than that of all Japanese municipalities on high-CII days (0.84). Therefore, the highest CII value occurred on the Pacific coast during summer with the condition of little local pollution.

Similar to the high-CII case, 30 days with the lowest CII average of all Japanese municipalities were selected (4/26, 5/29–6/2, 6/15, 6/16, 7/12, 7/15 in FY2014; 4/27, 5/13, 5/22, 5/27, 6/12, 6/13, 6/15, 7/31, 8/1, 8/2 in 2015; and 5/27, 5/28, 5/31, 6/10, 6/17, 6/18, 6/26, 6/27, 8/11, 9/1 in 2016). The average of CII values on these 30 low-CII days for each municipality are displayed in

**Table 3.** Same as Table 2 but for the average CII for the 30 high-CII days.

| Municipality | Prefecture | CII |
|---|---|---|
| Aogashima-mura (13402) | Tokyo | 0.902 |
| Hachijo-machi (13401) | Tokyo | 0.902 |
| Mikurajima-mura (13382) | Tokyo | 0.899 |
| Tsuno-cho (39411) | Kochi | 0.897 |
| Yusuhara-cho (39405) | Kochi | 0.897 |
| Kumano-shi (24212) | Mie | 0.897 |
| Kitayama-mura (30427) | Wakayama | 0.897 |
| Minabe-cho (30391) | Wakayama | 0.897 |
| Sakawa-cho (39402) | Kochi | 0.897 |
| Susaki-shi (39206) | Kochi | 0.897 |
| Average of all Japanese municipalites | | 0.836 |

Fig. 8 (d), and Table 4 shows the 10 municipalities with the highest average CII values on these days. These 10 municipalities located in southern remote islands, such as Miyakojima-shi in Okinawa Prefecture and Ogasawara-mura in Tokyo Prefecture. The average CII in these municipalities was 0.84–0.86, which was approximately 0.30–0.32, by CII, higher than that of all municipalities on low-CII days (0.54). The selected 30 days occurred especially at the end of spring and beginning of summer. Generally, the transboundary pollution effect is large in the cold season, and heavy local pollution occurs in summer because of

photochemical reactions induced by strong sunlight (e.g., Nagashima et al., 2010). These pollution effects are less pronounced in the remote islands, thus the CII maintained higher values.

We selected "Top 100 clean air cities" in Japan as one example of use in society of CII by the following method. The average of 30 highest daily CII values in the study period was calculated for each municipality. The 30 days were selected for each municipality, not as the case of Fig. 8 (c) and (d). Table 5 shows the 100 municipalities with the highest average CII values. The

municipalities in remote islands of Tokyo, around western Japan, especially around the Pacific coast, and Okinawa Prefectures, were selected.

## 4.2   Air cleanliness and human activities

Industrial activities, particularly fossil fuel combustion such as vehicles and power plants, are major sources of air pollutants, and air cleanliness is strongly related with human activities. In Sect. 4.2, we discuss the municipalities in Japan with not only

air cleanliness but also human activity. . We group them into four categories: 1) clean air with high human activity, 2) clean air with low human activity, 3) dirty air with high human activity, and 4) dirty air with low human activity. In this study, the common logarithm of population density ($n$), $\log_{10}(n)$, was employed to quantify human activities following e.g., Kerr and Currie (1995). The $n$ data were obtained from the 2015 Japanese national census (NSTAC, 2016). Figure 9 (a) shows the scatter

**Table 4.** Same as Table 2 but for the average CII for the 30 low-CII days.

| Municipality | Prefecture | CII |
|---|---|---|
| Miyakojima-shi (47214) | Okinawa | 0.860 |
| Ogasawara-mura (13421) | Tokyo | 0.857 |
| Tarama-son (47375) | Okinawa | 0.857 |
| Ishigaki-shi (47207) | Okinawa | 0.854 |
| Taketomi-cho (47381) | Okinawa | 0.854 |
| Minamidaito-son (47357) | Okinawa | 0.848 |
| Kitadaito-son (47358) | Okinawa | 0.845 |
| Yonaguni-cho (47382) | Okinawa | 0.841 |
| Kunigami-son (47301) | Okinawa | 0.838 |
| Higashi-son (47303) | Okinawa | 0.838 |
| Average of all Japanese municipalites | | 0.544 |

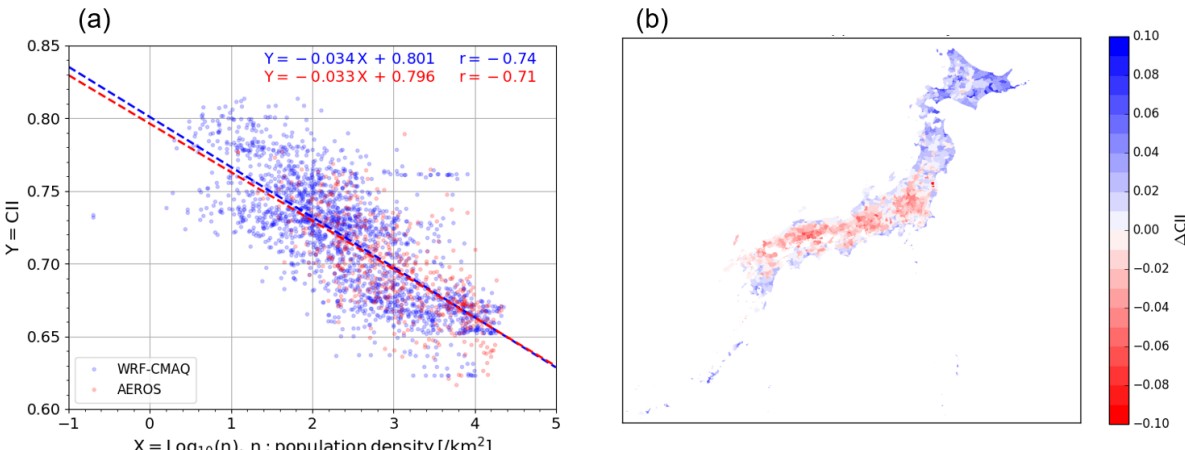

**Figure 9.** (a) Comparison of CII and population density ($n$) in each Japanese municipality. Dashed line shows linear regression of CII with $\log_{10}(n)$. Correlation coefficient ($r$) between CII values and $\log_{10}(n)$ is also shown in the upper right side. Blue and red color shows the WRF-CMAQ and AEROS results, respectively. (b) Distribution of differences in CII from the linear regression ($\Delta$CII) for the WRF-CMAQ model.

**Table 5.** "Top 100 clean air cities" in Japan. The municipal number is shown in parenthesis.

| Municipality | Prefecture |
| --- | --- |
| Nemuro-shi (1223), Kushiro-cho (1661), Akkeshi-cho (1662), Hamanaka-cho (1663) | Hokkaido |
| Niijima-mura (13363), Kozushima-mura (13364), Miyake-mura (13381), Mikurajima-mura (13382) | Tokyo |
| Hachijo-machi (13401), Aogashima-mura (13402), Ogasawara-mura (13421) | |
| Kaiyo-cho (36388) | Tokushima |
| Uwajima-shi (38203), Seiyo-shi (38214), Uchiko-cho (38422), Matsuno-cho (38484), Kihoku-cho (38488) | Ehime |
| Ainan-cho (38506) | |
| Aki-shi (39203), Tosa-shi (39205), Susaki-shi (39206), Sukumo-shi (39208), Tosashimizu-shi (39209) | Kochi |
| Shimanto-shi (39210), Toyo-cho (39301), Nahari-cho (39302), Tano-cho (39303), Yasuda-cho (39304) | |
| Kitagawa-mura (39305), Umaji-mura (39306), Geisei-mura (39307), Ino-cho (39386) | |
| Niyodogawa-cho (39387), Nakatosa-cho (39401), Sakawa-cho (39402), Ochi-cho (39403) | |
| Yusuhara-cho (39405), Hidaka-mura (39410), Tsuno-cho (39411), Shimanto-cho (39412) | |
| Otsuki-cho (39424), Mihara-mura (39427), Kuroshio-cho (39428) | |
| Taragi-machi (43505), Yunomae-machi (43506), Mizukami-mura (43507), Asagiri-cho (43514) | Kumamoto |
| Saiki-shi (44205) | Oita |
| Miyakonojyo-shi (45202), Nobeoka-shi (45203), Nichinan-shi (45204), Kobayashi-shi (45205) | Miyazaki |
| Hyuga-shi (45206), Kushima-shi (45207), Saito-shi (45208), Mimata-cho (45341), Kunitomi-cho (45382) | |
| Aya-cho (45383), Takanabe-cho (45401), Shintomi-cho (45402), Nishimera-son (45403) | |
| Kijyo-cho (45404), Kawaminami-cho (45405), Tsuno-cho (45406), Kadogawa-cho (45421) | |
| Morotsuka-son (45429), Shiiba-son (45430), Misato-cho (45431), Takachiho-cho (45441) | |
| Hinokage-cho (45442), Gokase-cho (45443) | |
| Kanoya-shi (46203), Makurazaki-shi (46204), Ibusuki-shi (46210), Nishinoomote-shi (46213) | Kagoshima |
| Soo-shi (46217), Kirishima-shi (46218), Shibushi-shi (46221), Amami-shi (46222) | |
| Minamikyushu-shi (46223), Osaki-cho (46468), Higashikushira-cho (46482) | |
| Kinko-cho (46490), Minamiosumi-cho (46491), Kimotsuki-cho (46492), Nakatane-cho (46501) | |
| Minamitane-cho (46502), Yakushima-cho (46505), Yamato-son (46523), Uken-son (46524) | |
| Setouchi-cho (46525), Tatsugo-cho (46527), Kikai-cho (46529) | |
| Miyakojima-shi (47214), Kunigami-son (47301), Higashi-son (47303), Minamidaito-son (47357) | Okinawa |
| Kitadaito-son (47358), Tarama-son (47375), Yonaguni-cho (47382) | |

plot of $\log_{10}(n)$ and average CII for the study period derived from the WRF-CMAQ model and the AEROS measurements for each municipality. A clear negative correlation between $\log_{10}(n)$ and the CII was observed and the $r$ values were $-0.74$ and $-0.71$ for WRF-CMAQ and AEROS, respectively. This negative correlation was formulated by the linear regression with the objective variable of CII and the explanatory variable of $\log_{10}(n)$, as shown by the dashed lines in Fig. 9 (a).

$$\text{Approximated CII} = a \times \log_{10}(n) + b \tag{4}$$

The parameters of $a$ and $b$ were estimated to be $-0.034 \pm 0.001$ and $0.801 \pm 0.002$ for WRF-CMAQ, and $-0.033 \pm 0.001$ and $0.796 \pm 0.005$ for AEROS, respectively. The negative correlation between $\log_{10}(n)$ and the CII value derived from WRF-CMAQ was reproduced from AEROS, and the parameters of $a$ and $b$ were agreed within their errors.

The CII value showed negative correlation with the human activity, thus the municipalities in groups 2 and 3 are in normal situation. The municipalities in group 1 is ideal case because such municipalities are expected to be industrially advanced as well as to succeed to maintain clean air environment. There are some issues in the municipalities in group 4 because such municipalities can not have clean air in spite of smaller population. It might indicate that there are large air pollution sources, such as large power plant, or air pollutants are transported from the outside. The degree of this categorizing is quantified by difference between the CII and the linear regression line, Eq. (4), ($\Delta$CII).

$$\Delta\text{CII} = \text{CII} - \text{Approximated CII} \tag{5}$$

The positive $\Delta$CII value means that the municipality is categorized in group 1, and the negative $\Delta$CII value does group 4. The distribution of $\Delta$CII in the average for the study period is shown in Fig. 9 (b), and Table 6 shows the 10 municipalities with the highest average $\Delta$CII values. All of these municipalities were in Hokkaido and Okinawa prefectures. The higher $\Delta$CII values were observed in northeastern Japan and coastal area. There are many industrial areas in western Japan (Li et al., 2017), which might be one reason for the lower $\Delta$CII values. A combination of CII and $\Delta$CII could be a useful way of evaluating air cleanliness in municipality.

## 5 Conclusions

We defined a novel concept of index for quantifying air cleanliness, namely CII. This index comprehensively evaluates the level of air cleanliness by normalizing the amounts of common air pollutants. A CII value of 1 indicates the absence of air pollutants, and 0 indicates that the amounts of air pollutants are the same as the normalization numerical criteria.

A model simulation was performed to visualize the air cleanliness of all 1896 municipalities in Japan using CII. We used $O_3$, SPM, $NO_2$, and $SO_2$ in this study, and their numerical environmental criteria were taken from the JEQS set by the MOE of Japan. The amounts of these species were calculated via the model combining the WRF model version 3.7 and CMAQ model version 5.1. The time period of the simulation was from 1 April 2014 to 31 March 2017, i.e., FY2014–2016. The CII values near the surface derived from the model were evaluated by comparing with those of the AEROS in situ observations, operated by the MOE of Japan. 498 municipalities were covered by the AEROS measurements. The difference of CII between WRF-CMAQ and AEROS was distributed in $0.00 \pm 0.02$ and $0.00 \pm 0.04$ for spatial and temporal bias, respectively. We concluded

**Table 6.** Ten municipalities with highest average $\Delta$CII value over the study period, FY2014–2016. The municipal number is shown in parenthesis.

| Municipality | Prefecture | $\Delta$CII | CII |
|---|---|---|---|
| Naha-shi (47201) | Okinawa | 0.095 | 0.762 |
| Urasoe-shi (47208) | Okinawa | 0.091 | 0.762 |
| Sapporo-shi, Shiroishi-ku (1104) | Hokkaido | 0.088 | 0.759 |
| Sapporo-shi, Chuo-ku (1101) | Hokkaido | 0.088 | 0.761 |
| Ginowann-shi (47205) | Okinawa | 0.088 | 0.762 |
| Sapporo-shi, Toyohira-ku (1105) | Hokkaido | 0.087 | 0.761 |
| Sapporo-shi, Higashi-ku (1103) | Hokkaido | 0.086 | 0.761 |
| Sapporo-shi, Kita-ku (1102) | Hokkaido | 0.086 | 0.761 |
| Tomigusuku-shi (47212) | Okinawa | 0.084 | 0.764 |
| Yonabaru-cho (47348) | Okinawa | 0.083 | 0.762 |
| Average of all Japanese municipalites | | 0.000 | 0.717 |

that the difference in CII derived from the WRF-CMAQ larger than 0.02 was significant to be reproduced by AEROS by averaging 30 values to reduce the temporal bias to be less than 0.01 ($\approx 0.04/\sqrt{30}$). Difference between CII and AQI was also discussed. The mean correlation coefficient ($r$) for the study period between WRF-CMAQ and AEROS was calculated for each municipality. The $r$ of CII and AQI was 0.66$\pm$0.05 (1 $\sigma$) and 0.57$\pm$0.06 (1 $\sigma$), respectively. The CII showed better agreement between WRF-CMAQ and AEROS than AQI because of the difference of definition between CII and AQI. The CII averages all normalized air pollutant amounts but the AQI employs only the maximum of the individual pollutants, i.e., any effects from the other air pollutants are ignored. This CII concept to comprehensively evaluate multiple air pollutants could be an advantage to quantify the air cleanliness.

Over the study period, FY2014–2016, the average CII value of Tokyo (23 wards), Seoul and Beijing was 0.67, 0.52 and 0.24, respectively. It means that the air in Tokyo was 1.5 and 2.3 times cleaner, i.e., less air pollutants, than those in Seoul and Beijing, respectively. The CII value varied spatially and temporally, corresponding to variations in $O_3$, SPM, $NO_2$, and $SO_2$. The municipalities in eastern Hokkaido Prefecture had the highest CII average values of approximately 0.81, which was 0.09, by CII, higher than the average values of all Japanese municipalities of 0.72. The extremely clean air with CII values, approximately 0.90, occurred in southern remote islands of Tokyo and around western the Pacific coast, i.e., Kochi, Mie and Wakayama Prefectures during summer with transport of unpolluted air from the ocean. The municipalities in southern remote islands in Okinawa and Tokyo Prefectures maintained their high CII values of 0.84–0.86, which was approximately 0.30–0.32, by CII, higher than that of all municipalities on low-CII days (0.54). Furthermore, the "Top 100 clean air cities" in Japan were presented as one example of how CII could be used in society.

We quantified the air cleanliness in municipality with respect to human industrial activities using population density. A negative correlation between CII and the population density was observed by both the WRF-CMAQ model and the AEROS measurement. The CII was approximated by a linear function of the common logarithm of population density. The differences of CII from this approximation line ($\Delta$CII) indicates the CII weighted by human activity. The municipalities with positive $\Delta$CII values are expected to maintain clean air and to be industrially advanced. Those with negative $\Delta$CII values are expected to have certain issues such as large air pollution source and air pollutants transported from the outside. A combination of CII and $\Delta$CII could be a useful way of evaluating air cleanliness in municipality.

The CII can be used in various scenarios, such as encouraging sightseeing and migration, investment and insurance company business, and city planning. The CII can be used for an advertisement of clean air for promoting sightseeing and migration for local governments. The CII is also effective to measure the potential of local brands and tourism resources. Private company can be expected to use CII for ESG (Environmental, Social and Governance) investment. If the CII could be associated with life expectancy, the CII can be applied to insurance business especially in Asian region where urban air pollution is a serious problem. City planning is also a possible use of CII because air cleanliness is related to urban form (e.g., McCarty and Kaza, 2015). As mentioned above, the CII has the potential to be applied to policy as well as company business in cities and countries around the world.

*Data availability.* The WRF-CMAQ model data in this publication can be accessed by contacting the authors. The AEROS measurement data are available through the following link: https://www.nies.go.jp/igreen. Japanese population density data are available through the following link: https://www.e-stat.go.jp/.

*Video supplement.* The CII daily mean for all 1896 Japanese municipalities is archived for each month over the study period, FY2014–2016.

*Author contributions.* Conceptualization, Leading by Y. K.; All authors; Model simulation, T. K.; Evaluation of data quality; T. O. S.; Manuscript writing, T. O. S. and T. K.; Writing significant contribution to paper, Y. K.; Review and editing, All authors.

*Competing interests.* The authors declare that they have no conflict of interest.

*Acknowledgements.* The WRF-CMAQ model simulation was performed by the computing system in the NICT Science cloud. We would like to thank the Big Data Analytics Laboratory of NICT and Suuri-Keikaku Co., Ltd. for supporting the computation. We gratefully acknowledge Iwao Hosako and Motoaki Yasui for their kind management of the research environment in NICT. We deeply appreciate Hideyuki Teraoka

in Ministry of Internal Affairs and Communications to give us an idea "TOP 100 clean air cities". TOS thanks to Seidai Nara for his polite technical support.

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
