# Peer review of "Novel index to comprehensively evaluate air cleanness: the "Clean aIr Index""

_Geoscience Communication, 2019_

## Referee Comment (RC1) · Anonymous Referee #1 · 2 Oct 2019

A. General comments:

In this manuscript, authors focused on their original index to evaluate air cleanness, named as CII. CII is defined as the difference from unity of the sum of relative cleanness of each pollutant (NO2, SO2, SPM and O3) normalized by each standard. This study is positioned as a basic research of their novel index and a demonstration for indicating reasonability and utility of the index. Such an index may be useful for capturing and understanding simply the air cleanness. Thus, the reviewer believes that this work has an important implication and is significant enough to be published in this journal. However, the present manuscript leaves several points to be improved, clarified, modified, and/or reconstructed, in order for readers to understand descriptions and to recognize the significance of this study clearly. Especially, it is necessary to

indicate more information and explanations with some arguments and/or references, and to clarify the story of discussion.

B. Important specific comments:

B1) Overall: uncertain and unclear points The referee thinks that the story which authors want to explain may be as follows: (1) validation of the model calculation (WRF/CMAQ) by comparison with observation data (AEROS), (2) explanation and interpretation of the novel cleanness index, CII, estimated for all the municipalities in Japan from the results of WRF/CMAQ, and (3) demonstration of the utility of the index, top 100 clean air cities in Japan. Would you please confirm that the referee's understanding is correct? Such a self-doubt of the referee is due to ambiguity and uncertainness in this study's aim, position, and significance, as follows, especially: (a) What are the differences among the indices? What are advantages to CII? Why CII, not AQI, for example? (b) What is the aim and significance of the top 100 clean air cities in Japan? (c) Descriptions and explanations on the methods and results are unclear. The major data in this study are based on the model calculation, aren't they? However, such critical points are not clearly found in the manuscript.

B2) Shallow descriptions without arguments and/or references: Some descriptions are without any arguments and/or references. The referee feels that authors say these descriptions definitively, without explanations.

Eg. Line 65: ..., because more than 90 - 95 % of Ox is composed of O3.

Eg. Line 136 (similar to Line 65)

Eg. Lines 176-: ..., because polluted air was transported from East Asia and ...

Eg. Lines 187-: (similar to Lines 176-)

Eg. Line 188-: typical lifetime of NO2

Eg. Line 251: There are many industrial areas in western Japan, ...

For example, are these the results of model calculation, or cited from references?

B3) Table 6 and Lines 225-228 Descriptions are only qualitative on the top 100 cities. What is the scientific implication?

B4) Line 239: Would you please indicate authors' opinion why 'except for Osaka' ? What is the situations in Osaka?

C. Other comments and Technical corrections:

C1) Explanations on model and method are insufficient. Essential parts of the model descriptions are not enough, the referee feels. For example: What is CMAQ? What reactions are considered? How to consider, for example, emission, deposition, secondary formation, transportation and diffusion, for SPM, O3, NO2, and SO2?

C2) The referee thinks that readers want to compare CII with other indices. Thus, please add explanations on other indices. For example, what is AQI? If possible, as a demonstration, please calculate AQI and compare some data and figures based on CII with those based on AQI.

C3) Line 137: What is 'Fig.3(a)' ?

C4) Line 165 and others: '1.2 times', for example, is not proper because CII is defined as '1 − (ratio of pollution)' in Eq.(1). In addition, the 'times' description is also not proper because the CII can be less than zero when the pollution is severe. Meanwhile, for the 'ratio of pollution', the 'multiple' description (like '1.2 times') is proper, because such 'ratio of pollution' is proportional to the quantities of pollutants. However, for the difference from unity in Eq.(1), such 'multiple' description cannot be explained to be proportional to some values.

C5) Line 165: Please indicate the sources of data and information on the pollutions in Seoul and Beijing.

C6) Figure 4: Would you please add the histogram of CII determined from AEROS data

and compare them with CII from CMAQ? Such additional comparisons can support the reasonability of CII from the model calculation.

C7) Overall: Would you please add the figures of CII determined from AEROS data (similar to Figs. 5, 6 and/or 7) and compare them with those from CMAQ? Such comparisons can support the advantages of CII from the model calculation. For example, 'figures acquired from AEROS data are insufficient but those from CMAQ are fine enough to discuss the spatial distributions and temporal variations of CIIs over Japan.'

C8) Overall: Geographical descriptions of Japan (eg. cities, prefectures, and islands) are insufficient. Readers unfamiliar with Japan cannot understand the information in this paper.

End of Comments.

---

## Referee Comment (RC2) · Kunihiko Arai (Referee) · 10 Oct 2019

[Overall summary] It is highly appreciated that the author defined "Clean Air Index = CII" as a new concept and aimed to apply it worldwide. The author has developed "CII: Air Cleanliness" for the first time in the world and proposed to set it as an international standard of air quality, which has been diverged in various countries until now. It contributes to environmental science in that air quality observation and future prediction can be done quantitatively. It also contributes to social aspects such as utilization in urban planning. "Delicious air" is of great interest in areas with severe environmental problems (especially China, Vietnam, Thailand, Indonesia, etc.). CII is an important factor for people moving abroad or staying longer. In addition, it is wonderful to open up the possibility of using CII to set a standard for incorporating "delicious air" as a tourism

resource. CII can also be highly evaluated for its potential to become a standard for tourism and migration.

[Comments and questions for the whole] ïČij The reliability of CII is not a problem because it uses the index set by WHO. ïČij Is there a correlation between the global distribution of CII and healthy life expectancy in each country? ïČij I think that it is too few to carry out model verification at 6 points. Why did it not be done at all points? ïČij Can you visualize the global distribution of CII in near real time? What tools do you need to do that? ïČij When creating CII for countries other than Japan, especially for emerging countries such as Africa and Southeast Asia, is there any data equivalent to that of the Japanese Ministry of the Environment? ïČij The goal of making CII as a global standard should be clearly stated as an issue for the future and written in the abstract. ïČij Please tell us why you normalized human activity in the population. Since this paper uses NO2 and SO2 for CII, I thought that the number of cars and the number of factories were more appropriate than the population density. ïČij Are there any plans to visualize the CII information on Web system in the future? Developing the system which can overlay CII with other information (disaster prevention and disaster prevention information app) and enable easy access to thematic map, e.g. land risk assessment, would be one of social implementations. ïČij Can you create CII for other countries with significant air pollution? For example, China, Southeast Asia, India, Nepal, Mongolia and Ulaanbaatar. ïČij Although there is a solid observation network in Japan, why do you use the model? Please write reason for needs of the model at the beginning of the appropriate chapter.

[Minor comments and questions] 25. Change a word, "Furthermore". In the sentence after "Furthermore", the reason for the change in air quality is written, so that it is not adequate. 26. Add references; why reduced labor productivity leads to increased demand for projected energy. 27. Why does GDP increase due to harvest loss derived from air pollution? How about excerpting one sentence from OECD2016? 28. How about excerpts from "McCarty and Kaza, 2015" about important issues in city plan-

ning? The reason for the change in air quality is written as "Increase in pollutants", but the reason for the importance of urban planning for air quality is not written. Therefore, the sentence balance in the paragraph is bad. 29-30. With regard to "clean water is", we insist on the necessity of creating an index based on "same as water", but is there a water world index? Provide references if any. If not, cut this sentence. 30. The meaning of "allow people to make more informed choices" is unknown. Please write specifically. 31. Easy access for citizens, easy to read, easy to understand, this is an important perspective for journals. This expression is written at the beginning of the sentence, and "Upgrading with experts and scientific data" will be described later. 35. Correct spelling. indexes or indices? 39. What are the selection criteria for that chemical? It is written a little in Chapter 2, what is the reason for making only 4? For example, is there a reference, whether it is a high rank, is it attracting attention in Japan, or is the standard that the country is most interested in? 40. I understood the meaning of "optimizing the numerical criteria" after reading Chapter 2. This means that the user can set any value. Since "optimizing" is likely to be misunderstood as an advanced optimization algorithm, it is expressed to avoid misunderstanding. 57. "O3, PM, NO2 and SO2 following the WHO AQG (WHO, 2005)" overlaps with Chapter 1. There is no need to erase. But write something already mentioned above, such as "mentioned above". 67. The health risks written in the introduction are also motivating research. Is it consistent with chemical substances SPM that are not health risks? 69. According to the cited document (1993), volcanic eruptions are said to have the highest SO2 emissions, but I hear that there is also a document that "the amount of sulfur supply to the atmosphere is more due to industrial activity than volcanic activity." Are there any recent papers, not 1993 references? 70. Regarding volcanoes, it is stated that SO2 emissions are high, and in line 110, it is stated that SO2 volcanic emissions were ignored in Japan, and there is a conflict. Furthermore, it is not consistent to include Kagoshima to evaluate the effects of volcanoes. Devise how to write. 89. Nudging is performed according to the 6-hour data. What is the time interval of the WRF-CMAQ calculation results? 100. Are the NOX, SO2, and SPM boundary conditions other than

O3 set in MOZART? 105. How did you find "the statistical secular changes in the annual total anthropogenic emissions"? Give a reference. 116. What is the reason for setting "R = 16km"? Is the domain grid interval related to 20km? 117. Outside of the domain such as Okinawa, it may not be necessary to consider CII. Evaluation is difficult because the scale is different. 130. As stated in "Volcanic emissions of SO2 were ignored (L110)", is it consistent with selecting Kagoshima because of the volcanoes? 134. It is written that the site of Sakurajima was excluded because it did not consider volcanoes in CMAQ, but are other sites in Kagoshima city susceptible to volcanoes? From Table 2, Kagoshima has a particularly poor correlation between NO2 and SO2. Is this the reason for the volcano? Or for reasons other than volcanoes? Did you enter Kagoshima to insist that the impact of the volcano is not so great? Clarify the intention to include Kagoshima. Or Kagoshima is not needed. Or let CMAQ consider Sakurajima's volcano. Do you have emission data for Sakurajima? 139. Correct the spelling. abovementioned-> above-mentioned 140. A good agreement with a correlation coefficient of 0.61 is a bit overstated. Is this a problem with the resolution and representativeness of the 10km model? 141. In Table 2, why is "CII" better in Akita and Nagano than in Kagoshima, where NO2 and SO2 are bad? 141. As with the time series, are the values in Table 2 a comparison of daily averages? 146. Put a dot after the formula number R. 146. Since it is a reaction by "hv", do the values in Table 2 and CII change depending on the presence of sunlight, that is, day and night? I think that the result of each day and night also has utility value (social needs). I think there is demand for people who need delicious air at noon and those who need it at night. 150. The reaction R3 causes the model to underestimate O3 and overestimate NO2, resulting in a poor correlation between O3 and NO2. Since CII is added together, it is offset and the correlation of CII does not deteriorate. Isn't it possible to properly devise an underestimation of O3 and an overestimation of NO2 in the model? And why does the correlation worsen in areas with few human origins such as Akita and Nagano? 153. Since the elimination of the NO2-O3 offset problem depends on the type of model, I think it will not be an advantage for all models. 157. There are things that look asym-

Interactive
comment

metric and those that don't. Devise how to write. 158. I think 1-$\sigma$ is a convention in this field. However, readers in other fields can easily misunderstand "-" as minus, and mistakenly read it as 1 minus $\sigma$. Isn't it just $\sigma$? 165. Which agency's data follows the denominator "s" for Seoul and Beijing's numerical criteria? 174. Write that the time being stated is around May. The writing style is unified. 185. Does "amount of O3 was relatively higher than the value of s" mean that x / s is larger than other spices? 187. The famous city name, Mega City, is written on the vertical axis in Figure 5. 193. In response to the above paragraph, it will not be "Consequently". It does not lead to cross-border pollution. How do you interpret Figure 5 to get evidence of cross-border pollution? I think there is cross-border pollution, but I can't interpret it from Figure 5 alone. 194. Since it overlaps with the 187th line of the upper paragraph, delete the sentence, "The variation in O3 had the most significant effect on seasonal variation in the CII. The spatial distribution of CII corresponded to those of NO2 and SO2." 195. The impact of domestic local sources can be seen in the vertical stripes in Figure 5, but there is insufficient evidence for "outside of Japan". 200. From Figure 6, it is difficult to tell the difference between good and bad places such as northern Japan. Devise the color scale to a palette of about 8 colors. 222. Add a reference to show that "Generally, the transboundary pollution effect" is significant in Japan in the spring. Write the reasons, such as the monsoon, or the high demand for coal-fired power generation in China in winter. 222. In the case of cross-border pollution, it is difficult to understand unless it is compared with a model such as PM2.5 that is expressed in time series. In addition, photochemical smog is a phenomenon under some very special circumstances in some areas, so it is better to expand the data representation a little more. That will be a future issue. 225. Is "The 30 highest daily mean CII values" shown in Fig. 6 (c)? 225. Based on the data in 6 prefectures in Japan, the municipalities in the prefecture are selected. However, from the nationwide data, there are naturally other regions with high value, so it is better to use these 6 cases. It may also be a good idea to list the seasons roughly. 232. Why is it "not fair" when it is fair to quantify CII on an objective basis? 245. Does normalization in human activity (population density)

mean to exclude the influence of human activity? Why is that? Is it for seeking potential cleanliness of the air? Want to see the impact of cross-border pollution? Write the reason and purpose at the beginning of the chapter. 250. Is it not just "neighboring municipality" but also transboundary pollution? For example, if the distribution of yellow sand and the distribution in Figure 7b overlap in previous studies, this is evidence of cross-border contamination. 282. Due to the circumstances of each individual, it is not necessary to strongly recommend moving to Hokkaido. Write about the causal relationship with healthy life expectancy, or write other reasons, such as clean air is better in nature and is more sustainable. However, just as people and factories set out to seek clean water, if people seek for clean air, they can put a load on clean nature and have the opposite effect. Sometimes it is better not to be a tourism business. 284. "enabled" is too much to say. Rather than saying that Korea and China alone can be applied to other countries, it is better to write that this method is simple and can be applied to countries and municipalities around the world.

---

## Referee Comment (RC3) · Kunihiko Arai (Referee) · 10 Oct 2019

[Impression]

(a) About tourism business The area where the starry sky is beautiful is a tourist spot. The Ministry of the Environment of Japan reports Achi Village in Nagano Prefecture as "a place suitable for observing Japan's starry sky". However, Achi Village is not ranked in the "Top 100 Municipal Rankings for Clean Air" in this study. An area with a beautiful starry sky can be a tourist attraction, but needs to be investigated to see if an area with beautiful air can become a tourist attraction. For example, in "sightseeing" or "business trips", the demand for cleanliness of air becomes clear by conducting interviews and questionnaires to people who want to go to a clean city. Needs surveys

such as questionnaire results will be a strong basis for claiming that CII is necessary for the tourism business.

(b) Insurance / real estate business In Southeast Asia such as Vietnam, Indonesia and Thailand, East Asia such as Mongolia and China, and South Asia such as India and Nepal, urban air pollution is severe. In cities and regions with severe air pollution, if the CII model can be used to set up medical insurance, it can be used for private use as evidence for insurance products. In some countries, the cause of death is air pollution. More certainty is required to use CII as an index for insurance companies. When considering foreign tourists (inbound), it can be used for indicators such as Japan x culture x nature x water and air. Persuasive power will increase if there are more specific data utilization cases. However, you need to be careful not to be criticized by the region.

(c) Corporate risk hedging The policy of increasing coal-fired power generation goes against the SDGs. In some cases, air pollution can lead to litigation issues. Dirty air can be a litigation risk for energy policies, power companies, construction companies, loan banks, etc. that have an environmental impact. In addition, these affiliates are at risk of being divested in ESG investments that are already spreading among investors. On the other hand, clean air is just an advertisement for local governments. Companies are also expected to invest ESG in activities that maintain and improve the clean air. CII is an effective index for measuring the potential of local brands and tourism resources. In countries and regions where there are few observation sites for air pollution, standardization of this CII model will lead to regional environmental assessment. In the future, it is possible that CII can be used as evidence for penal regulations for atmospheric environmental regulations in each urban area.

(d) Model expression ability In the future, the author expects to create not only Japan but also the global CII distribution. In that case, can the difference in seasonal change be correctly modeled in the mid-latitude and high-latitude zones, and in low-latitude zones, particularly in the rainforest, Indonesia, and the Amazon, forest fires, bushfire

haze, and volcanoes? I think that there is a lot of room for further study on whether such effects can be correctly incorporated into the model. The scope of this study is still within Japan. In the future, it will be necessary to verify in other regions whether it can be applied worldwide.

―――――――――――――――――――

---

## Referee Comment (RC4) · Anonymous Referee #3 · 21 Oct 2019

It is of great significance to develop the local and global air quality index for providing informative information to policy maker and citizen. The authors propose a simple index for qualifying air cleanness, "Clean aIr Index (CII)" and evaluate the air quality in Japan by using the CII. This work is challenging but the CII has critical problems for applying globally and locally. Additionally, the evaluation of CMAQ is too insufficient to analyze the air cleanness in Japan. This reviewer would recommend the publication of this manuscript after major revisions responding to following comments.

Major comments 1. The authors mentioned that "the purpose of the CII is to estimate the level of air cleanness that is not a health risk" (line 66). What is the "air cleanness" in this study? It should be explained the meaning of "air cleanness". The authors referred the WHO (2015) when they selected the pollutants in the CII. However, WHO (2015)

[Figure]

focused on the health effects of air pollution. As a result, the author's idea/concept about "air cleanness" is ambiguous.

2. The authors mentioned that "The CII can be used globally and locally by optimizing the numerical criteria". The author should explain how to set the value of numerical criteria when the CII is used globally. The air quality standards in each country are different due to the current status of air quality, health effects, socioeconomic and political aspects and other factors. Hence, the authors should propose the methodology for optimization of these differences.

3. As show in Table1, the averaging time of air quality standard for Ox (hourly) and other pollutants (SPM, $SO_2$ and $NO_2$; daily average) are different. How do the authors harmonize these differences?

4. The authors analyzed air cleanness in whole Japan by using the simulated results of CMAQ. However, the model evaluation is limited in only six cities. The CMAQ should be evaluated in all stations including remote sites. In particular, the municipalities in Hokkaido and Okinawa which are selected as those with highest CII value in Chapter 4 should be included in the model evaluation.

5. The authors mentioned that "The model underestimates the amount of $O_3$ and overestimates that of $NO_2$ in case of large contribution of the reaction (R3), i.e., NO titration effect." (lines 149-150). Is this correct? If the model can reproduce well the NO titration effect, there are less discrepancies between model and observation. In general, the regional chemical transport model such as CMAQ tends to be underestimate the NO titration in urban area because the model cannot reflect the effects of local emissions. Additionally, the CMAQ tends to overestimate the $O_3$ concentration in Tokyo (For example, see Akimoto et al., 2019) . (Ref.) Akimoto et al., Atmos. Chem. Phys., 19, 603–615, 2019 https://doi.org/10.5194/acp-19-603-2019

Minor comments 1. Line 67: "The amount of SPM was simply assumed as [SPM] = ([PM10] + [PM2.5])/2 in this study" should be moved to section 3.2 because this

assumption may be applied in the conversion of PM10 and PM2.5 of CMAQ to SPM.

2. Lines 163-166ïïjŽ Is it appropriate to analyze the air quality in Seoul and Beijing by using the CII based on the Japanese's standards?

3. Lines 249-251: In "The (delta)CII value reflects the transport of air pollutants from around the municipality rather than the CII value", what is the meaning of negative value of (delta)CII?
* * *

---

## Author Comment (AC1) · 13 Dec 2019

Dear Anonymous Referee #1

General comments from Referee: In this manuscript, authors focused on their original index to evaluate air cleanness, named as CII. CII is defined as the difference from unity of the sum of relative cleanness of each pollutant ($NO_2$, $SO_2$, SPM and $O_3$) normalized by each standard. This study is positioned as a basic research of their novel index and a demonstration for indicating reasonability and utility of the index. Such an index may be useful for capturing and understanding simply the air cleanness. Thus, the reviewer believes that this work has an important implication and is significant enough to be published in this journal. However, the present manuscript leaves several points to be

improved, clarified, modified, and/or reconstructed, in order for readers to understand descriptions and to recognize the significance of this study clearly. Especially, it is necessary to indicate more information and explanations with some arguments and/or references, and to clarify the story of discussion.

Author's response: We greatly appreciate your efforts to help us improve our manuscript. Yes, this manuscript is fundamental, and the aim is topropose our concept of "Clean aIr Index, CII." We answered your valuable comments point by point as the attached files, especially for the abstract, introduction and conclusion to state the objective of CII more clearly. We hope that our manuscript is suitable for publication in GC.

Sincerely yours,

Tomohiro Sato National Institute of Information and Communications Technology

Please also note the supplement to this comment:
https://www.geosci-commun-discuss.net/gc-2019-16/gc-2019-16-AC1-supplement.pdf
* * *
[Figure]

**Supplement:**

**Point-By-Point Reply to Referee Comment 1 from Anonymous Referee #1**

**Comment from Referee:**

**B. Important specific comments:** B1) Overall: uncertain and unclear points The referee thinks that the story which authors want to explain may be as follows: (1) validation of the model calculation (WRF/CMAQ) by comparison with observation data (AEROS), (2) explanation and interpretation of the novel cleanness index, CII, estimated for all the municipalities in Japan from the results of WRF/CMAQ, and (3) demonstration of the utility of the index, top 100 clean air cities in Japan. Would you please confirm that the referee's understanding is correct? Such a self-doubt of the referee is due to ambiguity and uncertainness in this study's aim, position, and significance, as follows, especially: (a) What are the differences among the indices? What are advantages to CII? Why CII, not AQI, for example? (b) What is the aim and significance of the top 100 clean air cities in Japan? (c) Descriptions and explanations on the methods and results are unclear. The major data in this study are based on the model calculation, aren't they? However, such critical points are not clearly found in the manuscript.

**Author's response:**

Yes, your understanding is right. We answered your question (a), (b) and (c) as follows. (a) As you mentioned, the statement about CII was unclear and we added a statement of AQI in Sect. 1. The advantage of CII is to represent the combined effects of multiple pollutants. A comparison study between CII and AQI using WRF-CMAQ and AEROS was added in Sect. 3.3, and please also read the Author's response to C2. We also improved the description of AQI in introduction. (b) We believe the CII has many possibilities to be used in society. We presented "Top 100 clean air cities" as one example of use in society of CII not for scientific meaning. (c) Descriptions of our model simulation in Sect. 3.1 was updated following your comments. Please also read the Author's response to C1.

**Author's changes in the manuscript:**

Page 2 Line 44 – 50, Page 11 Sect. 3.3

Page 1 Line 23 – 24, Page 17 Line 320 – 321, Page 22 Line 404

Page 5 Sect 3.1

**Comment from Referee:**

B2) Shallow descriptions without arguments and/or references: Some descriptions are without any arguments and/or references. The referee feels that authors say these descriptions definitively,

without explanations.

Eg. Line 65: : : : , because more than 90 - 95 % of Ox is composed of O3.

Eg. Line 136 (similar to Line 65)

Eg. Lines 176-: : : :, because polluted air was transported from East Asia and : : :

Eg. Lines 187-: (similar to Lines 176-)

Eg. Line 188-: typical lifetime of NO2

Eg. Line 251: There are many industrial areas in western Japan, : : :

For example, are these the results of model calculation, or cited from references?

**Author's response:**

We added the following references in the manuscript.

· Akimoto, H.: Overview of policy actions and observational data for $PM_{2.5}$ and $O_3$ in Japan: A study of urban air quality improvement in Asia, (2017).

· Park, M. E. et al.: New approach to monitor transboundary particulate pollution over Northeast Asia. Atmos. Chem. Phys., 14(2), 659-674, (2014).

· Kenagy, H. S. et al.: NOx lifetime and NOy partitioning during WINTER. J. Geophys. Res. Atmos., 123(17), 9813-9827, 2018.

· Li, M. et al.,: MIX: a mosaic Asian anthropogenic emission inventory under the international collaboration framework of the MICS-Asia and HTAP. Atmos. Chem. Phys., 17(2), 935-963, 2017.

**Author's changes in the manuscript:**

Page 4 Line 79, Page 7 Line 188

Page 14 Line 264, 276, 278

Page 21 Line 366

**Comment from Referee:**

B3) Table 6 and Lines 225-228 Descriptions are only qualitative on the top 100 cities. What is the scientific implication?

**Author's response:**

Thank you for pointing it out. We believe the CII has many possibilities to be used in society, and "Top 100 clean air cities" was presented as one example of use in society of CII not for scientific meaning.

**Author's changes in the manuscript:**

Page 1 Line 23 – 24, Page 17 Line 320 – 321, Page 22 Line 404

**Comment from Referee:**

B4) Line 239: Would you please indicate authors' opinion why 'except for Osaka'? What is the situations in Osaka?

**Author's response:**

Discrepancy between WRF-CMAQ and AEROS in Osaka in the previous manuscript was much improved by the following revisions. We changed the x value of $O_3$ from the daily average to the maximum of hourly value in 24 hours to be consistent between the time-span of x and that of s, following the comment from Anonymous Referee #3 (Major comment 3). Also, following your comment (C7), we investigated correlation between CII and population density (n) for all Japanese municipalities, where the AEROS observation covered more than 80% of the days of the study period, including Osaka. Both WRF-CMAQ and AEROS showed negative correlation between CII and population density, and the linear regression of CII (= Y) with $\log_{10}(n)$ (= X) was consistent within their errors as follows. The slope was -0.034±0.001 and -0.033±0.001 for WRF-CMAQ and AEROS, respectively. The intercept was 0.801±0.002 and 0.796±0.005 for WRF-CMAQ and AEROS, respectively. Therefore, we updated Fig. 7 and the description of CII and the human activity in Sect. 4.3 in the previous manuscript.

**Author's changes in the manuscript:**

Page 11 Sect. 3.3, Page 17 Sect. 4.3, Page 18 Fig. 9

**Comment from Referee:**

**C. Other comments and Technical corrections:** C1) Explanations on model and method are insufficient. Essential parts of the model descriptions are not enough, the referee feels. For example: What is CMAQ? What reactions are considered? How to consider, for example, emission, deposition, secondary formation, transportation and diffusion, for SPM, O3, NO2, and SO2?

**Author's response:**

CMAQ is a chemical transport model and we used CMAQ to calculate air pollutant amounts for this study. WRF is used for a calculation of meteorological fields (e.g., temperature, wind, and

humidity) which is essential for accurate calculation of air pollutant amounts. We added the descriptions of emission, deposition, secondary formation, transportation and diffusion in Sect. 3.1 following your suggestion.

**Author's changes in the manuscript:**
Page 5 Sect 3.1

**Comment from Referee:**
C2) The referee thinks that readers want to compare CII with other indices. Thus, please add explanations on other indices. For example, what is AQI? If possible, as a demonstration, please calculate AQI and compare some data and figures based on CII with those based on AQI.

**Author's response:**
Thank you so much for this comment. We additionally calculated the AQI value based on our WRF-CMAQ model and the AEROS measurements. The correlation coefficients (r) of daily CII value for the study period between WRF-CMAQ and AEROS were compared with those of AQI. Figure 5 in the revised manuscript shows the distribution of r of (a) CII and (b) AQI. The mean of r for all municipalities, where the AEROS observation covered more than 80% of the days of the study period, was $0.664 \pm 0.048$ (1 $\sigma$) and $0.566 \pm 0.056$ (1 $\sigma$) for CII and AQI, respectively. The CII showed better agreement between WRF-CMAQ and AEROS than AQI. This is because of the difference of calculation method. In the calculation of AQI, only the air pollutant that causes the largest health risk is taken into account and the other air pollutants are ignored. In the calculation of CII, all of air pollutants, $O_3$, SPM, $NO_2$ and $SO_2$, are averaged with normalization by their numerical standards. The definition of CII relatively cancels discrepancies in each species in case that the amounts reciprocally vary as $O_3$ and $NO_2$ by the chemical reaction; $NO + O_3 \rightarrow NO_2 + O_2$. Therefore, the cancellation of discrepancy in individual species in the definition of CII is a significant advantage for quantifying air cleanness using the proposed model.

**Author's changes in the manuscript:**
Page 2 Line 44 – 50, Page 11 Sect. 3.3

**Comment from Referee:**
C3) Line 137: What is 'Fig.3(a)' ?

**Author's response:**

Thank you for pointing it out. We corrected this typo.

**Author's changes in the manuscript:**

Page 7 Line 190

**Comment from Referee:**

C4) Line 165 and others: '1.2 times', for example, is not proper because CII is defined as '1 – (ratio of pollution)' in Eq.(1). In addition, the 'times' description is also not proper because the CII can be less than zero when the pollution is severe. Meanwhile, for the 'ratio of pollution', the 'multiple' description (like '1.2 times') is proper, because such 'ratio of pollution' is proportional to the quantities of pollutants. However, for the difference from unity in Eq.(1), such 'multiple' description cannot be explained to be proportional to some values.

**Author's response:**

We appreciate your comment and agree with your suggestion. The value of 1 – CII is proportional to air pollutants amounts, thus we improved the statement as "The average and standard deviation (1 σ) of CII was 0.67±0.10, 0.52±0.18, and 0.24±0.32 in Tokyo, Seoul, and Beijing, respectively. The value of 1 - CII is proportional to air pollutant amounts, and the air in Tokyo was 1.5 and 2.3 times cleaner, less air pollutant amounts, than those in Seoul and Beijing, respectively."

**Author's changes in the manuscript:**

Page 1 Line 17 – 20, Page 12 Line 252 – 253, Page 22 Line 393 – 396

**Comment from Referee:**

C5) Line 165: Please indicate the sources of data and information on the pollutions in Seoul and Beijing.

**Author's response:**

The data of the pollutants in Seoul and Beijing are derived from our CMAQ model results, as those cities are inside the Domain 1. We added the description in the end of Sect. 3.1.

**Author's changes in the manuscript:**

Page 7 Line 167 – 168

**Comment from Referee:**

C6) Figure 4: Would you please add the histogram of CII determined from AEROS data and compare them with CII from CMAQ? Such additional comparisons can support the reasonability of CII from the model calculation.

**Author's response:**

Yes, we added the histogram of the CII difference between WRF-CMAQ and AEROS for all Japanese municipalities.

**Author's changes in the manuscript:**

Page 11 Fig. 5

**Comment from Referee:**

C7) Overall: Would you please add the figures of CII determined from AEROS data (similar to Figs. 5, 6 and/or 7) and compare them with those from CMAQ? Such comparisons can support the advantages of CII from the model calculation. For example, 'figures acquired from AEROS data are insufficient but those from CMAQ are fine enough to discuss the spatial distributions and temporal variations of CIIs over Japan.'

**Author's response:**

We performed a comparison study for all AEROS observation sites following your comment, and we discussed the spatial and temporal bias in our model simulation by statistical approach as follows. 498 in 1896 municipalities were covered by the AEROS measurements and the statistical method could be possible by including all AEROS observation sites to cover large number of samples. We deeply appreciate your valuable comment. To investigate the spatial bias between municipalities in our model simulation, we compared the CII mean of all days in the study period for each municipality between WRF-CMAQ and AEROS. The mean and standard deviation (1 sigma) of CII difference (WRF-CMAQ - AEROS) were 0.000 and 0.022, respectively. In the similar way, we investigated the daily temporal bias by comparing the CII mean of all Japanese municipalities for each day between WRF-CMAQ and AEROS. The mean and standard deviation (1 $\sigma$) of CII difference were 0.000 and 0.044, respectively. We averaged the CII values for at least 30 days to compare the CII value among municipalities to reduce the temporal bias in CII difference between WRF-CMAQ and AEROS to be less than 0.01. Consequently, we regarded

that the CII difference larger than 0.02 is significant.

**Author's changes in the manuscript:**

Page 7 Sect. 3.2, Page 8 Fig. 2

**Comment from Referee:**

C8) Overall: Geographical descriptions of Japan (eg. cities, prefectures, and islands) are insufficient. Readers unfamiliar with Japan cannot understand the information in this paper.

**Author's response:**

Thank you so much for pointing it out. We showed the location of the municipalities mentioned in this manuscript.

**Author's changes in the manuscript:**

Page 9 Fig. 3, Page 10 Line 206 – 207

[revised manuscript text omitted]

---

## Author Comment (AC2) · 13 Dec 2019

Dear Referee Kunihiko Arai

Overall summary from Referee: It is highly appreciated that the author defined "Clean Air Index = CII" as a new concept and aimed to apply it worldwide. The author has developed "CII: Air Cleanliness" for the first time in the world and proposed to set it as an international standard of air quality, which has been diverged in various countries until now. It contributes to environmental science in that air quality observation and future prediction can be done quantitatively. It also contributes to social aspects such as utilization in urban planning. "Delicious air" is of great interest in areas with severe environmental problems (especially China, Vietnam, Thailand, Indonesia, etc.). CII

is an important factor for people moving abroad or staying longer. In addition, it is wonderful to open up the possibility of using CII to set a standard for incorporating "delicious air" as a tourism resource. CII can also be highly evaluated for its potential to become a standard for tourism and migration.

Author's response: We greatly appreciate your efforts to help us improve our manuscript. Yes, CII has many possibilities in social aspects such as urban planning, residence, tourism and migration. We believe the clean air is as valuable a resource as clean water and this manuscript would work to make the CII concept common in the world. We answered your valuable comments point by point as the attached files. We hope that our manuscript is suitable for publication in GC.

Sincerely yours,

Tomohiro Sato National Institute of Information and Communications Technology

Please also note the supplement to this comment:
https://www.geosci-commun-discuss.net/gc-2019-16/gc-2019-16-AC2-supplement.pdf

**Supplement:**

**Point-By-Point Reply to Referee Comment 2 from Referee Kunihiko Arai**

**Comment from Referee:**

**[Comments and questions for the whole]** The reliability of CII is not a problem because it uses the index set by WHO. Is there a correlation between the global distribution of CII and healthy life expectancy in each country? I think that it is too few to carry out model verification at 6 points. Why did it not be done at all points? Can you visualize the global distribution of CII in near real time? What tools do you need to do that? When creating CII for countries other than Japan, especially for emerging countries such as Africa and Southeast Asia, is there any data equivalent to that of the Japanese Ministry of the Environment? The goal of making CII as a global standard should be clearly stated as an issue for the future and written in the abstract.

**Author's response:**

Thank you so much for your valuable comments. We performed a comparison study for all AEROS observation sites following your comment. 498 in 1896 municipalities were covered by the AEROS measurements. The statements and figures for validation of WRF-CMAQ using the AEROS measurements were updated in the revised manuscript. The global distribution of CII can be derived using global model such as GEOS-chem [Wang et al., 2004] and CHASER [Sudo et al., 2002a; 2002b]. We recommend to use the WHO Air Quality Guidelines for the numerical criteria for the global distribution of CII because it is the only criteria for air pollutants defined by the international organization as far as we know. The WHO AQG is also employed for applying CII to countries with no environmental standards. Following your comments, we improved the abstract and introduction to clearly state our aim of CII to make a global standard for the air cleanness.

- Sudo, K., Takahashi, M., Kurokawa, J. I., & Akimoto, H. (2002a). CHASER: A global chemical model of the troposphere 1. Model description. *Journal of Geophysical Research: Atmospheres*, *107*(D17), ACH-7.
- Sudo, K., Takahashi, M., & Akimoto, H. (2002b). CHASER: A global chemical model of the troposphere 2. Model results and evaluation. *Journal of Geophysical Research: Atmospheres*, *107*(D21), ACH-9.
- Wang, Y. X., McElroy, M. B., Jacob, D. J., & Yantosca, R. M. (2004). A nested grid formulation for chemical transport over Asia: Applications to CO. *Journal of Geophysical Research: Atmospheres*, *109*(D22).

**Author's changes in the manuscript:**

Page 1 Line 3, Page 2 Line 51 – 52, Page 4 Line 79 – 81, Page 7 Sect. 3.2,

**Comment from Referee:**

**[Comments and questions for the whole]** Please tell us why you normalized human activity in the population. Since this paper uses NO2 and SO2 for CII, I thought that the number of cars and the number of factories were more appropriate than the population density.

**Author's response:**

Yes, the number of cars and the number of factories are also suitable for this research. But we could not find such a database that covers all 1896 municipalities in the study period (FY2014-2016). The common logarithm of population density showed good correlation with $NO_2$ (r = 0.80) and $SO_2$ (r = 0.74). The population density might not be the best, but appropriate to quantify the human activity.

**Comment from Referee:**

**[Comments and questions for the whole]** Are there any plans to visualize the CII information on Web system in the future? Developing the system which can overlay CII with other information (disaster prevention and disaster prevention information app) and enable easy access to thematic map, e.g. land risk assessment, would be one of social implementations.

**Author's response:**

We had no such plans but we would like to adopt your idea in the future. Thank you for your nice suggestion.

**Comment from Referee:**

**[Comments and questions for the whole]** Can you create CII for other countries with significant air pollution? For example, China, Southeast Asia, India, Nepal, Mongolia and Ulaanbaatar.

**Author's response:**

Yes, we can derive the CII values for other countries by using a global model, such as GEOS-chem [Wang et al., 2004] and CHASER [Sudo et al., 2002a; 2002b]. The WHO AQG standards should be employed as the numerical criteria in CII in case of no environmental standards in the other countries.

- Sudo, K., Takahashi, M., Kurokawa, J. I., & Akimoto, H. (2002a). CHASER: A global chemical model of the troposphere 1. Model description. *Journal of Geophysical Research:*

*Atmospheres*, *107*(D17), ACH-7.

- Sudo, K., Takahashi, M., & Akimoto, H. (2002b). CHASER: A global chemical model of the troposphere 2. Model results and evaluation. *Journal of Geophysical Research: Atmospheres*, *107*(D21), ACH-9.
- Wang, Y. X., McElroy, M. B., Jacob, D. J., & Yantosca, R. M. (2004). A nested grid formulation for chemical transport over Asia: Applications to CO. *Journal of Geophysical Research: Atmospheres*, *109*(D22).

**Comment from Referee:**

**[Comments and questions for the whole]** Although there is a solid observation network in Japan, why do you use the model? Please write reason for needs of the model at the beginning of the appropriate chapter.

**Author's response:**

We used the model because the AEROS measurements do not cover the all 1896 municipalities.

**Author's changes in the manuscript:**

Page 4 Line 99 – 100

**Comment from Referee:**

**[Minor comments and questions]** 25. Change a word, "Furthermore". In the sentence after "Furthermore", the reason for the change in air quality is written, so that it is not adequate. 26. Add references; why reduced labor productivity leads to increased demand for projected energy. 27. Why does GDP increase due to harvest loss derived from air pollution? How about excerpting one sentence from OECD2016?

**Author's response:**

We revised this sentence because it was too long and partly wrong as follows.

**Author's changes in the manuscript:**

Page 2 Line 29 – 34

**Comment from Referee:**

**[Minor comments and questions]** 28. How about excerpts from "McCarty and Kaza, 2015"

about important issues in city planning? The reason for the change in air quality is written as "Increase in pollutants", but the reason for the importance of urban planning for air quality is not written. Therefore, the sentence balance in the paragraph is bad.

**Author's response:**

Yes, we agree with your suggestion. This sentence was isolated in the paragraph. We moved this statement from introduction to conclusion.

**Author's changes in the manuscript:**

Page 23 Line 423

**Comment from Referee:**

**[Minor comments and questions]** 29-30. With regard to "clean water is", we insist on the necessity of creating an index based on "same as water", but is there a water world index? Provide references if any. If not, cut this sentence.

**Author's response:**

Yes, there is a water world index, "Global Drinking Water Quality Index (GDWQI)." We added this reference in the manuscript.

**Author's changes in the manuscript:**

Page 2 Line 35 – 36

**Comment from Referee:**

**[Minor comments and questions]** 30. The meaning of "allow people to make more informed choices" is unknown. Please write specifically.

**Author's response:**

We modified this sentence as follows.

**Author's changes in the manuscript:**

Page 2 Line 37

**Comment from Referee:**

**[Minor comments and questions]** 31. Easy access for citizens, easy to read, easy to understand, this is an important perspective for journals. This expression is written at the beginning of the sentence, and "Upgrading with experts and scientific data" will be described later.

**Author's response:**

We agreed with your suggestion and modified this sentence as follows.

**Author's changes in the manuscript:**

Page 2 Line 37 – 39

**Comment from Referee:**

**[Minor comments and questions]** 35. Correct spelling. indexes or indices?

**Author's response:**

Thank you for pointing it out. We corrected the term "indexes" to "indices" as follows.

**Author's changes in the manuscript:**

Page 2 Line 40, 41, 42

**Comment from Referee:**

**[Minor comments and questions]** 39. What are the selection criteria for that chemical? It is written a little in Chapter 2, what is the reason for making only 4? For example, is there a reference, whether it is a high rank, is it attracting attention in Japan, or is the standard that the country is most interested in?

**Author's response:**

These 4 pollutants are selected from the WHO Air Quality Guidelines which is most common guideline for air quality as far as we know.

**Comment from Referee:**

**[Minor comments and questions]** 40. I understood the meaning of "optimizing the numerical criteria" after reading Chapter 2. This means that the user can set any value. Since "optimizing" is likely to be misunderstood as an advanced optimization algorithm, it is expressed to avoid

misunderstanding.

**Author's response:**

We agreed with your suggestion and changed from "optimizing" to "defining."

**Author's changes in the manuscript:**

Page 4 Line 79 – 80

**Comment from Referee:**

**[Minor comments and questions]** 57. "O3, PM, NO2 and SO2 following the WHO AQG (WHO, 2005)" overlaps with Chapter 1. There is no need to erase. But write something already mentioned above, such as "mentioned above".

**Author's response:**

We agreed with your suggestion and added "as mentioned above" in the manuscript.

**Author's changes in the manuscript:**

Page 3 Line 73

**Comment from Referee:**

**[Minor comments and questions]** 67. The health risks written in the introduction are also motivating research. Is it consistent with chemical substances SPM that are not health risks?

**Author's response:**

There are many studies to report association between SPM and health risk, such as Ueda et al., (2010).

- Ueda, K., Nitta, H., & Odajima, H. (2010). The effects of weather, air pollutants, and Asian dust on hospitalization for asthma in Fukuoka. *Environmental health and preventive medicine*, *15*(6), 350.

**Comment from Referee:**

**[Minor comments and questions]** 69. According to the cited document (1993), volcanic eruptions are said to have the highest SO2 emissions, but I hear that there is also a document that

"the amount of sulfur supply to the atmosphere is more due to industrial activity than volcanic activity." Are there any recent papers, not 1993 references? 70. Regarding volcanoes, it is stated that SO2 emissions are high, and in line 110, it is stated that SO2 volcanic emissions were ignored in Japan, and there is a conflict. Furthermore, it is not consistent to include Kagoshima to evaluate the effects of volcanoes. Devise how to write.

**Author's response:**
We deeply appreciate for pointing it out. As you mentioned, the description of our $SO_2$ calculation was ambiguous. Major source of $SO_2$ emission in Japan is combustion of fossil fuels [Wakamatsu et al., 2013]. Amount of $SO_2$ occasionally rise because of volcanic eruption, but only in a short period of volcanic eruption. In this study, the $SO_2$ numerical criterion is for the daily average, and the CII values are compared by averaging at least 30 days values. This process dilutes temporal $SO_2$ increase due to volcanic eruption. That is why we ignored $SO_2$ emission in our model simulation. We modified the statements about $SO_2$ calculation as follows to make this point clearer.

**Author's changes in the manuscript:**
Page 4 Lines 92 – 95
Page 6 Lines 150 – 153

**Comment from Referee:**
**[Minor comments and questions]** 89. Nudging is performed according to the 6-hour data. What is the time interval of the WRF-CMAQ calculation results?

**Author's response:**
The time interval of both WRF and CMAQ outputs is 1 hour. We added the description about the time interval.

**Author's changes in the manuscript:**
Page 4 Line 104

**Comment from Referee:**
**[Minor comments and questions]** 100. Are the NOX, SO2, and SPM boundary conditions other than O3 set in MOZART?

**Author's response:**

Yes. MOZART provided the distributions of more than 80 kinds of chemical species and aerosols, including NOX, SO2, PM and O3, as added in the manuscript.

**Author's changes in the manuscript:**

Page 6 Line 158 – 159

**Comment from Referee:**

[**Minor comments and questions**] 105. How did you find "the statistical secular changes in the annual total anthropogenic emissions"? Give a reference.

**Author's response:**

We added the reference in the manuscript.

·   Crippa, M., Oreggioni, G., Guizzardi, D., Muntean, M. Schaaf, E., Lo Vullo, E., Solazzo, E., Monforti-Ferrario, F., Olivier, J., and Vignati, E.: Fossil CO2 and GHG emissions of all world countries, https://doi:10.2760/687800, 2019.

**Author's changes in the manuscript:**

Page 6 Line 146

**Comment from Referee:**

[**Minor comments and questions**] 116. What is the reason for setting "R = 16km"? Is the domain grid interval related to 20km?

**Author's response:**

It is due to the convenience of the derivation of air quality at the municipal office, to be able to refer at least 1 model grid point. Theoretically the necessary smallest value of R is 14.1 km ($\sqrt{2}*10$ km), so we consider R=16 km is a good definition.

**Comment from Referee:**

[**Minor comments and questions**] 117. Outside of the domain such as Okinawa, it may not be necessary to consider CII. Evaluation is difficult because the scale is different.

**Author's response:**

In Okinawa, there are no large local emission sources and major source of pollution is transboundary effects from outside. Transboundary pollution was well reproduced with larger scale. That is why we ignored difference of scale in two domains.

**Comment from Referee:**

[**Minor comments and questions**] 130. As stated in "Volcanic emissions of SO2 were ignored (L110)", is it consistent with selecting Kagoshima because of the volcanoes? 134. It is written that the site of Sakurajima was excluded because it did not consider volcanoes in CMAQ, but are other sites in Kagoshima city susceptible to volcanoes? From Table 2, Kagoshima has a particularly poor correlation between NO2 and SO2. Is this the reason for the volcano? Or for reasons other than volcanoes? Did you enter Kagoshima to insist that the impact of the volcano is not so great? Clarify the intention to include Kagoshima. Or Kagoshima is not needed. Or let CMAQ consider Sakurajima's volcano. Do you have emission data for Sakurajima?

**Author's response:**

In our simulation the effects of volcanic activities are not considered because of the reason described in Author's response to Minor comments and questions for Line 69 and 70. Also, we changed the strategy of validation of our WRF-CMAQ calculation from specific case study with 6 cities to statistical approach with all the AEROS sites. We drastically changed the statements of the comparison study in Sect. 3.2.

**Author's changes in the manuscript:**

Page 7 Sect. 3.2

**Comment from Referee:**

[**Minor comments and questions**] 139. Correct the spelling. abovementioned-> above-mentioned

**Author's response:**

Thank you for pointing it out. But this sentence was removed because of changing the validation strategy from specific case study with 6 cities to statistical approach with all the AEROS sites.

**Comment from Referee:**

**[Minor comments and questions]** 140. A good agreement with a correlation coefficient of 0.61 is a bit overstated. Is this a problem with the resolution and representativeness of the 10km model?

**Author's response:**

Major cause of discrepancy between our WRF-CMAQ model and the AEROS measurements is probably local and unpredictable emissions. In our model, local emissions from industries, such as traffics and combustion, were given by the MIX inventories. The temporal resolution of the MIX inventory is monthly, thus the daily or hourly-scale emissions could not be reproduced by our model. That is why we compared the CII values by averaging at least 30 days.

**Comment from Referee:**

**[Minor comments and questions]** 141. In Table 2, why is "CII" better in Akita and Nagano than in Kagoshima, where NO2 and SO2 are bad? 141. As with the time series, are the values in Table 2 a comparison of daily averages?

**Author's response:**

We could not understand your point. Do you mean why CII in Akita and Nagano (r = 0.61) were worse than that in Kagoshima (r = 0.65)? But this table was removed because of changing the validation strategy from specific case study with 6 cities to statistical approach with all the AEROS sites.

**Comment from Referee:**

**[Minor comments and questions]** 146. Put a dot after the formula number R.

**Author's response:**

We appreciate your comment. We confirmed the style defined in the Copernicus Publications, and "(R1)" is correct, not "(R.1)". We unified the style to "(R1)" in the manuscript.

**Author's changes in the manuscript:**

Page 14 Line 281

**Comment from Referee:**

**[Minor comments and questions]** 146. Since it is a reaction by "hv", do the values in Table 2

and CII change depending on the presence of sunlight, that is, day and night? I think that the result of each day and night also has utility value (social needs). I think there is demand for people who need delicious air at noon and those who need it at night.

**Author's response:**

Yes, air pollutant amounts vary in day and night as you suggested. But CII defined in this paper can only derived to daily unit because the we used the numerical criteria for 24-hour average.

**Comment from Referee:**

**[Minor comments and questions]** 150. The reaction R3 causes the model to underestimate O3 and overestimate NO2, resulting in a poor correlation between O3 and NO2. Since CII is added together, it is offset and the correlation of CII does not deteriorate. Isn't it possible to properly devise an underestimation of O3 and an overestimation of NO2 in the model? And why does the correlation worsen in areas with few human origins such as Akita and Nagano?

**Author's response:**

We appreciate your comment. Adjusting amounts of $O_3$ and $NO_2$ is theoretically possible but quite hard because many parameters, such as convergence rate, are intricately related with each other in the model. Although further investigation is required, one possible reason for worse correlation of CII in Akita and Nagano can be an uncertainty in natural emission sources.

**Comment from Referee:**

**[Minor comments and questions]** 153. Since the elimination of the NO2-O3 offset problem depends on the type of model, I think it will not be an advantage for all models.

**Author's response:**

We appreciate your comment and agree with your suggestion. Following the comment from Anonymous Referee 1, we compared CII and AQI using both WRF-CMAQ and AEROS. The correlation between WRF-CMAQ and AEROS was better in CII than AQI. This is due to difference of the calculation method, i.e., CII averages normalized air pollutant amounts, but AQI only employs the highest air pollutant and the others are ignored. This is an advantage of CII to comprehensively estimate all air pollutants and we stated it in the revised manuscript.

**Author's changes in the manuscript:**

Page 22 Line 390 – 391

**Comment from Referee:**

**[Minor comments and questions]** 157. There are things that look asymmetric and those that don't. Devise how to write.

**Author's response:**

This figure was removed because of changing the validation strategy from specific case study with 6 cities to statistical approach with all the AEROS sites.

**Comment from Referee:**

**[Minor comments and questions]** 158. I think 1-σ is a convention in this field. However, readers in other fields can easily misunderstand "-" as minus, and mistakenly read it as 1 minus σ. Isn't it just σ?

**Author's response:**

We changed the term "1-σ" to "1 σ".

**Author's changes in the manuscript:**

Page 9 Caption in Fig. 4, Page 10 Line 220, 223, Page 11 Caption in Fig. 5, Line 232, Page 12 Line 249, Page 15 Caption in Fig. 8, Line 290, 292, Page 22 Line 384

**Comment from Referee:**

**[Minor comments and questions]** 165. Which agency's data follows the denominator "s" for Seoul and Beijing's numerical criteria?

**Author's response:**

The same numerical criteria for Japan are used for Seoul and Beijing to directly compare the CII values of Seoul, Beijing and Japanese municipalities.

**Comment from Referee:**

**[Minor comments and questions]** 174. Write that the time being stated is around May. The

writing style is unified.

**Answer from authors**

We modified the statement as below.

**Author's changes in the manuscript:**

Page 14 Line 261

**Comment from Referee:**

**[Minor comments and questions]** 185. Does "amount of O3 was relatively higher than the value of s" mean that x / s is larger than other spices?

**Answer from authors**

Yes, you are correct. The sentence was improved as below. Thank you pointing it out.

**Author's changes in the manuscript:**

Page 14 Line 273 – 274

**Comment from Referee:**

**[Minor comments and questions]** 187. The famous city name, Mega City, is written on the vertical axis in Figure 5.

**Author's response:**

We added Sapporo, Tokyo, Yokohama, Nagoya, Osaka, Kobe, and Fukuoka on the left of the vertical axis in the figures.

**Author's changes in the manuscript:**

Page 8 Fig. 2, Page 13 Fig. 7

**Comment from Referee:**

**[Minor comments and questions]** 193. In response to the above paragraph, it will not be "Consequently". It does not lead to cross-border pollution. How do you interpret Figure 5 to get evidence of cross-border pollution? I think there is cross-border pollution, but I can't interpret it

from Figure 5 alone. 194. Since it overlaps with the 187th line of the upper paragraph, delete the sentence, "The variation in O3 had the most significant effect on seasonal variation in the CII. The spatial distribution of CII corresponded to those of NO2 and SO2." 195. The impact of domestic local sources can be seen in the vertical stripes in Figure 5, but there is insufficient evidence for "outside of Japan".

**Author's response:**

Thank you so much for pointing it out. We wrote this paragraph to summarize dominant source of spatial and temporal distribution of CII, but it should be done in the conclusion section. We removed this paragraph.

**Comment from Referee:**

[**Minor comments and questions**] 200. From Figure 6, it is difficult to tell the difference between good and bad places such as northern Japan. Devise the color scale to a palette of about 8 colors.

**Author's response:**

Yes, we agree with your suggestion that the color resolution was too detailed. We analyzed that the difference in CII derived from the WRF-CMAQ larger than 0.02 was significant to be reproduced by AEROS by averaging 30 values. Thus, we changed the color resolution to 0.02 grid in this figure.

**Author's changes in the manuscript:**

Page 15 Fig. 8

**Comment from Referee:**

[**Minor comments and questions**] 222. Add a reference to show that "Generally, the transboundary pollution effect" is significant in Japan in the spring. Write the reasons, such as the monsoon, or the high demand for coal-fired power generation in China in winter. 222. In the case of cross-border pollution, it is difficult to understand unless it is compared with a model such as PM2.5 that is expressed in time series. In addition, photochemical smog is a phenomenon under some very special circumstances in some areas, so it is better to expand the data representation a little more. That will be a future issue.

**Author's response:**

There are many previous studies of source of air pollutants in Japan using chemical transport models. As you mentioned, this topic is quite important for the environmental policy and should be further investigated in the future. We improved the sentence and added the following reference.

·   Nagashima, T., Ohara, T., Sudo, K., & Akimoto, H. (2010). The relative importance of various source regions on East Asian surface ozone. *Atmospheric Chemistry and Physics*, *10*(22), 11305-11322.

**Author's changes in the manuscript:**
Page 17 Line 317 – 318

**Comment from Referee:**
**[Minor comments and questions]** 225. Is "The 30 highest daily mean CII values" shown in Fig. 6 (c)?

**Author's response:**
Thank you for pointing it out. "Top 100 clean air cities" was selected by different way. The 30 days with highest daily CII values were selected for each municipality. But for Fig. 6 (c), the 30 highest days were selected for the average CII for all 1896 municipalities. We selected "Top 100 clean air cities" by quite simple way in this manuscript but the selection method should be civilized through discussion in the future. We improved the description as follows.

**Author's changes in the manuscript:**
Page 17 Line 321 – 322

**Comment from Referee:**
**[Minor comments and questions]** 225. Based on the data in 6 prefectures in Japan, the municipalities in the prefecture are selected. However, from the nationwide data, there are naturally other regions with high value, so it is better to use these 6 cases. It may also be a good idea to list the seasons roughly.

**Author's response:**
We deeply apologize that we could not understand meaning of your question. The "Top 100 clean air cities" were selected using all 1896 municipalities data not only the six, Akita, Tokyo, Nagano, Osaka, Fukuoka and Kagoshima cities. It would be so nice if you could give us more detailed

explanation.

**Comment from Referee:**

**[Minor comments and questions]** 232. Why is it "not fair" when it is fair to quantify CII on an objective basis? 245. Does normalization in human activity (population density) mean to exclude the influence of human activity? Why is that? Is it for seeking potential cleanliness of the air? Want to see the impact of cross-border pollution? Write the reason and purpose at the beginning of the chapter. 250. Is it not just "neighboring municipality" but also transboundary pollution? For example, if the distribution of yellow sand and the distribution in Figure 7b overlap in previous studies, this is evidence of cross-border contamination.

**Author's response:**

Thank you so much for pointing it out. Our objective to normalize CII with human activity is categorizing municipalities into four groups; 1) Clean air with high human activity, 2) Clean air with low human activity, 3) Dirty air with high human activity, and 4) Dirty air with low human activity. The CII value showed negative correlation with the human activity, thus the municipalities in groups 2 and 3 are in normal situation. The municipalities in group 1 is ideal case because such municipalities are expected to be industrially advanced as well as to succeed to maintain clean environment. Problems are in the municipalities in group 4, because only few people live in but the environment can not be saved. It means that there are large air pollution sources such as large power plant or air pollutants are transported from the outside. We guess this interpretation is also important to apply the CII concept in social usage such as residence and environmental policies. We improved this section to make this point clearer as follows.

**Author's changes in the manuscript:**

Page 17 Line 330 – Page 20 Line 333

**Comment from Referee:**

**[Minor comments and questions]** 282. Due to the circumstances of each individual, it is not necessary to strongly recommend moving to Hokkaido. Write about the causal relationship with healthy life expectancy, or write other reasons, such as clean air is better in nature and is more sustainable. However, just as people and factories set out to seek clean water, if people seek for clean air, they can put a load on clean nature and have the opposite effect. Sometimes it is better not to be a tourism business.

**Author's response:**

Yes, we agree with your suggestion. We removed this sentence.

**Comment from Referee:**

**[Minor comments and questions]** 284. "enabled" is too much to say. Rather than saying that Korea and China alone can be applied to other countries, it is better to write that this method is simple and can be applied to countries and municipalities around the world.

**Author's response:**

We appreciate your valuable comments. We improved the sentence following your suggestion.

**Author's changes in the manuscript:**

Page 23 Line 424 – 425

[revised manuscript text omitted]

---

## Author Comment (AC3) · 13 Dec 2019

**Point-By-Point Reply to Referee Comment 3 from Referee Kunihiko Arai**

**Comment from Referee:**

[Impression] (a) About tourism business

The area where the starry sky is beautiful is a tourist spot. The Ministry of the Environment of Japan reports Achi Village in Nagano Prefecture as "a place suitable for observing Japan's starry sky". However, Achi Village is not ranked in the "Top 100 Municipal Rankings for Clean Air" in this study. An area with a beautiful starry sky can be a tourist attraction, but needs to be investigated to see if an area with beautiful air can become a tourist attraction. For example, in "sightseeing" or "business trips", the demand for cleanliness of air becomes clear by conducting interviews and questionnaires to people who want to go to a clean city. Needs surveys such as questionnaire results will be a strong basis for claiming that CII is necessary for the tourism business.

**Author's response:**

We deeply appreciate your encourages. Thank you so much for showing us clear vision to implement CII to the tourism business. We guess that the reason why Achi Village in Nagano Prefecture was not selected in "Top 100 clean air cities" is that CII is not an index to quantify the air visibility. SPM is most effective for the air visibility but daily mean value of SPM was used in this study (not focusing on the nighttime). CII for the sky visibility, cloud condition should be included, could be one option of social use of CII.

**Comment from Referee:**

(b) Insurance / real estate business

In Southeast Asia such as Vietnam, Indonesia and Thailand, East Asia such as Mongolia and China, and South Asia such as India and Nepal, urban air pollution is severe. In cities and regions with severe air pollution, if the CII model can be used to set up medical insurance, it can be used for private use as evidence for insurance products. In some countries, the cause of death is air pollution. More certainty is required to use CII as an index for insurance companies. When considering foreign tourists (inbound), it can be used for indicators such as Japan x culture x nature x water and air. Persuasive power will increase if there are more specific data utilization cases. However, you need to be careful not to be criticized by the region.

**Author's response:**

CII is a simple index, and can be applied to other countries where the air pollution is more severe

than Japan. If we compare the CII values between country and others, the numerical criteria should be given by the WHO Air Quality Guidelines because it is the only criteria for air pollutants defined by the international organization as far as we know. Also if we could set reliable standards for health risk, CII for medical insurance could be made. Analysis of association with CII and social data such as number of insurance client or foreign tourist number would be helpful to implement CII to the insurance and foreign tourist businesses.

**Comment from Referee:**

(c) Corporate risk hedging

The policy of increasing coal-fired power generation goes against the SDGs. In some cases, air pollution can lead to litigation issues. Dirty air can be a litigation risk for energy policies, power companies, construction companies, loan banks, etc. that have an environmental impact. In addition, these affiliates are at risk of being divested in ESG investments that are already spreading among investors. On the other hand, clean air is just an advertisement for local governments. Companies are also expected to invest ESG in activities that maintain and improve the clean air. CII is an effective index for measuring the potential of local brands and tourism resources. In countries and regions where there are few observation sites for air pollution, standardization of this CII model will lead to regional environmental assessment. In the future, it is possible that CII can be used as evidence for penal regulations for atmospheric environmental regulations in each urban area.

**Author's response:**

We deeply appreciate that you could understand our research. This is our motivation to present the CII concept in this manuscript. We added the perspective of the CII in the business scene in the last paragraph of the conclusion.

**Author's changes in the manuscript:**

Page 23 Line 419 – 425

**Comment from Referee:**

(d) Model expression ability

In the future, the author expects to create not only Japan but also the global CII distribution. In that case, can the difference in seasonal change be correctly modeled in the mid-latitude and high-latitude zones, and in low-latitude zones, particularly in the rainforest, Indonesia, and the Amazon,

forest fires, bushfire haze, and volcanoes? I think that there is a lot of room for further study on whether such effects can be correctly incorporated into the model. The scope of this study is still within Japan. In the future, it will be necessary to verify in other regions whether it can be applied worldwide.

**Author's response:**

Yes, evaluation of air cleanness in the world should be the scope of this CII research in the future. According to the report of WHO, approximately 3.7 million deaths in the world were caused by exposure to the air pollution in 2012, and Asia is especially severe. As you mentioned, modeling such a local pollution is a scientific problem to overcome. Validation strategy is also important because the observation data near the surface for whole of the world is required. Satellite measurement is useful for global coverage but is difficult to extract the surface data from space, especially for the surface ozone. Thank you so much for giving us such a positive feedback and it would be so nice if you could continue to discuss with us.

[revised manuscript text omitted]

---

## Author Comment (AC4) · 13 Dec 2019

Dear Anonymous Referee #3

Overall comment from Referee: It is of great significance to develop the local and global air quality index for providing informative information to policy maker and citizen. The authors propose a simple index for qualifying air cleanness, "Clean aIr Index (CII)" and evaluate the air quality in Japan by using the CII. This work is challenging but the CII has critical problems for applying globally and locally. Additionally, the evaluation of CMAQ is too insufficient to analyze the air cleanness in Japan. This reviewer would recommend the publication of this manuscript after major revisions responding to following comments.

Author's response: We greatly appreciate your efforts to help us improve our manuscript. We answered your valuable comments point by point as the attached files. We improved abstract and introduction to state the purpose of this study, i.e., CII, and also a validation of our WRF-CMAQ calculation was much improved by comparing with the all AEROS observation sites. We hope that our manuscript is suitable for publication in GC.

Sincerely yours,

Tomohiro Sato National Institute of Information and Communications Technology

Please also note the supplement to this comment:
https://www.geosci-commun-discuss.net/gc-2019-16/gc-2019-16-AC4-supplement.pdf

[Figure]

**Supplement:**

**Point-By-Point Reply to Referee Comment 4 from Anonymous Referee 3**

**Comment from Referee:**

**Major comments 1.** The authors mentioned that "the purpose of the CII is to estimate the level of air cleanness that is not a health risk" (line 66). What is the "air cleanness" in this study? It should be explained the meaning of "air cleanness". The authors referred the WHO (2015) when they selected the pollutants in the CII. However, WHO (2015) focused on the health effects of air pollution. As a result, the author's idea/concept about "air cleanness" is ambiguous.

**Author's response:**

Thank you so much for pointing it out. As you mentioned, the statement about "air cleanness" was unclear and this sentence was misleading. The purpose of this manuscript is to propose the concept of CII to make globally common standard for air quality because the presented worldwide used Air Quality Index (AQI) has critical problem that is not applicable to multi-pollutant air pollution. We modified the sentence that you mentioned as well as the abstract and introduction to state our purpose more clearly.

**Author's changes in the manuscript:**

Page 1 Line 3 – 4, Page 2 Line 35 – 52

**Comment from Referee:**

**Major comments 2.** The authors mentioned that "The CII can be used globally and locally by optimizing the numerical criteria". The author should explain how to set the value of numerical criteria when the CII is used globally. The air quality standards in each country are different due to the current status of air quality, health effects, socioeconomic and political aspects and other factors. Hence, the authors should propose the methodology for optimization of these differences.

**Author's response:**

We suggest the WHO Air Quality Guidelines for the numerical criteria for the global distribution of CII because it is the only criteria for air pollutants defined by the international organization as far as we know. We added this statement as follows.

**Author's changes in the manuscript:**

Page 4 Line 79 – 81

**Comment from Referee:**

**Major comments 3.** As show in Table1, the averaging time of air quality standard for Ox (hourly) and other pollutants (SPM, SO2 and NO2; daily average) are different. How do the authors harmonize these differences?

**Author's response:**

We deeply appreciate your valuable comment. In the previous manuscript, we used daily average of $O_3$ as x and the numerical criterion of Ox hourly limit as s, ignoring time difference between x and s. We changed x of $O_3$ as maximum of hourly value in 24 hours to be consistent between x and s. All CII values, figures, and tables were updated in the current manuscript.

**Author's changes in the manuscript:**

Page 1 Line 18, 19, 23, Page 3 Table 1, Page 3 Line 75 – Page 4 Line 78, Page 8 Fig. 2, Page 9 Fig. 4, Page 10 Line 221, 223, Page 11 Fig. 5, Line 225, 232, Page 12 Fig. 6, Line 246, 249 – 252, Page 13 Fig. 7, Page 14 Line 261, 263, Page 15 Fig. 8, Line 290, 292, Page 16 Table 2, Line 294 – 295, 298 – 300, 307, 310 – 311, Page 17 Table 3, Line 315, 316, Page 18 Table 4, Fig. 9, Page 19 Table 5, Page 20 Line 337, 340, 342 – 343, Page 21 Table 6, Line 382, Page 22 Line 385, 387, 394 – 395, 397 – 399, 402 – 403, 409

**Comment from Referee:**

**Major comments 4.** The authors analyzed air cleanness in whole Japan by using the simulated results of CMAQ. However, the model evaluation is limited in only six cities. The CMAQ should be evaluated in all stations including remote sites. In particular, the municipalities in Hokkaido and Okinawa which are selected as those with highest CII value in Chapter 4 should be included in the model evaluation.

**Author's response:**

We performed a comparison study for all AEROS observation sites following your comment, and we discussed the spatial and temporal bias in our model simulation by statistical approach as follows. 498 in 1896 municipalities were covered by the AEROS measurements and the statistical method could be possible by including all AEROS observation sites to cover large number of samples. We deeply appreciate your valuable comment. To investigate the spatial bias between municipalities in our model simulation, we compared the CII mean of all days in the study period for each municipality between WRF-CMAQ and AEROS. The mean and standard deviation (1

sigma) of CII difference (WRF-CMAQ - AEROS) were 0.000 and 0.022, respectively. In the similar way, we investigated the daily temporal bias by comparing the CII mean of all Japanese municipalities for each day between WRF-CMAQ and AEROS. The mean and standard deviation (1 σ) of CII difference were 0.000 and 0.044, respectively. We averaged the CII values for at least 30 days to compare the CII value among municipalities to reduce the temporal bias in CII difference between WRF-CMAQ and AEROS to be less than 0.01. Consequently, we regarded that the CII difference larger than 0.02 is significant.

**Author's changes in the manuscript:**
Page 7 Sect. 3.2

**Comment from Referee:**
**Major comments 5.** The authors mentioned that "The model underestimates the amount of O3 and overestimates that of NO2 in case of large contribution of the reaction (R3), i.e., NO titration effect." (lines 149-150). Is this correct? If the model can reproduce well the NO titration effect, there are less discrepancies between model and observation. In general, the regional chemical transport model such as CMAQ tends to be underestimate the NO titration in urban area because the model cannot reflect the effects of local emissions. Additionally, the CMAQ tends to overestimate the O3 concentration in Tokyo (For example, see Akimoto et al., 2019).
(Ref.) Akimoto et al., Atmos. Chem. Phys., 19, 603–615, 2019 https://doi.org/10.5194/acp-19-603-2019

**Author's response:**
We deeply appreciate your valuable comments and agree that our manuscript was quite misleading. Considering the comment from Referee #1 to compare CII and AQI, we improved the statements as follows.

**Author's changes in the manuscript:**
Page 11 Sect. 3.3

**Comment from Referee:**
**Minor comments 1.** Line 67: "The amount of SPM was simply assumed as [SPM] = ([PM10] + [PM2.5])/2 in this study" should be moved to section 3.2 because this assumption may be applied in the conversion of PM10 and PM2.5 of CMAQ to SPM.

**Author's response:**

We agree with your suggestion. The sentence was moved to Sect. 3.

**Author's changes in the manuscript:**

Page 4 Line 107

**Comment from Referee:**

**Minor comments 2.** Lines 163-166: Is it appropriate to analyze the air quality in Seoul and Beijing by using the CII based on the Japanese's standards?

**Author's response:**

Yes, the same numerical criteria for Japan should be used for Seoul and Beijing to directly compare the CII values of Seoul, Beijing and Japanese municipalities.

**Comment from Referee:**

**Minor comments 3.** Lines 249-251: In "The (delta)CII value reflects the transport of air pollutants from around the municipality rather than the CII value", what is the meaning of negative value of (delta)CII?

**Author's response:**

We stated the purpose of this analysis using $\Delta$CII more clearly in Sect. 4.3. Our objective to introduce $\Delta$CII by normalizing CII with human activity is categorizing municipalities into four groups; 1) Clean air with high human activity, 2) Clean air with low human activity, 3) Dirty air with high human activity, and 4) Dirty air with low human activity. The negative $\Delta$CII value means the municipality is categorized in group 4. There might be some issues in this group because only few people live in but the environment can not be saved. It means 
[revised manuscript text omitted]

---

## Referee Report (RR1)

[referee-annotated manuscript omitted]

---

## Author Response (AR2)

**Reply to Editor**

**Comment from the editor**

Editor Decision: Publish subject to minor revisions (further review by editor) (02 Jul 2020) by Sam Illingworth

Comments to the Author:

Thank you for engaging so throughly with the peer review process, and my apologies for the length of time that this has taken, due to circumstances that were sadly beyond our control. Having now throughly reviewed the manuscript and the referee' comments, we will be delighted to publish this work in Geoscience Communication, once a few small edits have been made, as requested by Referee 2. The referee report from Referee 2 contains several suggestions to improve the readability of the paper, please could address all of these and upload a revised manuscript. Once this has been done I will ensure that I expedite the final editorial review.

Thank you again for your continued patience and professionalism, and I look forward to seeing the updated manuscript.

Many Thanks,

Sam

**Author's response**

Dear Prof. Illingworth

We greatly appreciate your great effort to improve our manuscript. We answered all comments from the Referee 2 point-by-point as follows. We hope that our manuscript is suitable for publication in GC.

Sincerely yours,

Tomohiro Sato, Yasuko Kasai

National Institute of Information and Communications Technology

**Point-By-Point Reply to Report #2 from Anonymous Referee #4**

**Comment from Referee:**
**A. General comments from Referee:**
This is an interesting study in terms of defining a new straightforward and simple methodology to calculate an index for assessing clean air (CII), which takes account of the four main pollutants. The authors use a model of air pollution over Japan and an alternative index to assess which better matches ground measurements. The CII can also be used in a relative manner by changing the normalisation factor to account for each country's legal framework. The authors apply the CII to the pollution in Japan over a number of years to provide maps and statistics of air quality in the region.

Please see the attached pdf for comments and corrections.

Referee Report: gc-2019-16-referee-report.pdf

**Author's response:**
Thank you so much for your quite careful reading. We greatly appreciate your efforts to help us improve our manuscript. We answered your valuable comments point by point as follows. We are sure that our manuscript could be more readable and understandable than the previous one. We hope that our manuscript is suitable for publication in GC.

**Comment from Referee:**
Title: cleanliness is better than cleanness e.g. see
https://www.learnersdictionary.com/qa/cleanliness-and-cleanness

**Author's response:**
We appreciate your comment. We changed all terms "cleanness" to "cleanliness" following your suggestion.

**Author's changes in the manuscript:**
Title, Page 1 Line 2, 3, 12, 13, 24, Page 2 Line 33, 47, 48, 49, 56, 57, Page 3 Line 74, Page 11 Line 213, 214, Page 13 Line 222, 223, Page 16 Line 257, 259, 260, Page 18 Line 285, 287, 288, 290, Page 19 Line 304, 315, Page 20 Line 322, 328

**Comment from Referee:**

P1 Line 15: what do you mean by averaging?

**Author's response:**

Thank you for pointing it out. It means that the mean of correlation coefficients for 498 municipalities was 0.66±0.05. We improved the sentence as follows.

**Author's changes in the manuscript:**

Page 1 Lines 15 – 18

"The CII values calculated by the WRF-CMAQ model and the AEROS measurements showed good agreement with a correlation coefficient of 0.66±0.05, averaging 498 municipalities where the AEROS measurements have operated, which was higher than that of Air Quality Index (AQI) of 0.57±0.06."

→

"The CII values calculated by the WRF-CMAQ model and the AEROS measurements showed good agreement. The mean of correlation coefficient for the CII values of 498 municipalities where the AEROS measurements have operated was 0.66±0.05, which was higher than that of Air Quality Index (AQI) of 0.57±0.06."

**Comment from Referee:**

Page 2 Line 30: assuming that the energy comes from fossil fuels?

**Author's response:**

Yes, the combustion of fossil fuels is one of the largest sources of air pollutants emission. We improved the sentence as follows.

**Author's changes in the manuscript:**

Page 2 Line 30

"power generation" → "power generation using fossil fuels"

**Comment from Referee:**

Page 10 Figure 4: Do you mean frequency?
Page 10 Figure 4: I don't understand the difference between the histograms

**Author's response:**

Thank you for pointing it out. We showed the histograms by the probability density function, not

by the frequency, in the previous manuscript. But we changed Figures 4 and 5 to the histograms of the frequency following your suggestion because the frequency is more popular than the probability density function as far as we surveyed.

The difference between the panels (a) and (b) in Fig. 4 is that the panel (a) shows the spatial bias of the CII difference between WRF-CMAQ and AEROS, and the panel (b) shows the temporal bias. We improved the statements and Figure 4 as follows.

**Author's changes in the manuscript:**

Page 9 Lines 184 - 186

"We compared the CII mean of all days in the study period between WRF-CMAQ and AEROS for each municipality to investigate the spatial bias in Fig. 4 (a)."

→

"We compared the CII mean of all days in the study period between WRF-CMAQ and AEROS for each municipality. The CII difference for each municipality is shown in Fig. 4 (a) to investigate the spatial bias."

Page 9 Lines 189 – 191

"In the similar way, we investigated the daily temporal bias by comparing the CII mean of all Japanese municipalities for each day."

→

"In the similar way, we compared the CII mean of all Japanese municipalities for each day during the study period to investigate the daily temporal bias. Figure 4 (b) shows the histogram of the CII difference for each day."

Page 10 Figure 4

[Figure]

[Figure]

→

[Figure]

Page 11 Figure 5

→

**Comment from Referee:**

Page 10 Lines 197 – 198: what other indices?

**Author's response:**

We appreciate you comment. We regarded AQI as representative of indices to evaluate the air quality. We improved the statement as follows.

**Author's changes in the manuscript:**
Page 10 Lines 197 – 198
"the other indices" → "indices to evaluate the air quality"

**Comment from Referee:**
Page 16 Line 260: We group them into four categories:

**Author's response:**
Thank you for pointing it out. We followed your suggestion.

**Author's changes in the manuscript:**
Page 16 Line 260
", i.e.," → ". We group them into four categories:"

**Other corrections from Referee:**
Referee Report: gc-2019-16-referee-report.pdf

**Author's response:**
We deeply appreciate your careful reading and many corrections. We improved our manuscript as follows.

**Author's changes in the manuscript:**
Page 1 Lines 2 – 3: "deserves a valuable" → "is as valuable a"
Page 1 Line 3: "Global" → "A global"
Page 1 Line 5: Add "a set of"
Page 1 Line 10: "obvious" → "fixed"
Page 1 Line 11: Remove "namely", "given" → ", as directed"
Page 1 Line 12: Add "data from"
Page 2 Line 53: "obvious" → "fixed"
Page 6 Line 135: "used CMAQ model had" → "CMAQ model used"
Page 6 Line 152: "is" → "are"

Page 10 Caption of Fig. 4: "Their" → "The"

Page 10 Line 197: Add "the"

Page 10 Line 199: "The correlation coefficient (r) of mean" → "The mean of correlation coefficient (r)"

Page 10 Line 204: "pollutant that causes" → "pollutant that cause", "is" → "are"

Page 10 Line 205: "all of" → "four"

Page 11 Line 212: "with" → "by"

Page 14 Line 235: Add "all", Remove "selected"

Page 14 Line 241: "few" → "little"

Page 18 Line 276: "save" → "have"

Page 19 Table 6: "-0.000" → "0.000"

Page 19 Line 299: "The correlation coefficient (r) of mean" → "The mean correlation coefficient (r)"

Page 19 Lines 302 – 303: "all individuals" → "the individual pollutants"

Page 19 Line 306: Remove "amounts of"

Page 19 Line 313: Add "the"

Page 19 Lines 313 – 314: "was" → "were"

Page 19 Line 314: "CII to" → "how CII could"

Page 19 Lines 315 – 316: ". Population density was used to quantify human activities in this study" → "using population density"

Page 20 Line 319: "normalized" → "weighted"

Page 20 Line 329: "a" → "the"

[revised manuscript text omitted]